# UNDERSTANDING HINDSIGHT GOAL RELABELING REQUIRES RETHINKING DIVERGENCE MINIMIZATION

## ABSTRACT

Hindsight goal relabeling has become a foundational technique for multi-goal reinforcement learning (RL). The idea is quite simple: any arbitrary trajectory can be seen as an expert demonstration for reaching the trajectory's end state. Intuitively, this procedure trains a goal-conditioned policy to imitate a sub-optimal expert. However, this connection between imitation and hindsight relabeling is not well understood. Modern imitation learning algorithms are described in the language of divergence minimization, and yet it remains an open problem how to recast hindsight goal relabeling into that framework. In this work, we develop a unified objective for goal-reaching that explains such a connection, from which we can derive goal-conditioned supervised learning (GCSL) and the reward function in hindsight experience replay (HER) from first principles. Experimentally, we find that despite recent advances in goal-conditioned behaviour cloning (BC), multi-goal $Q$-learning can *still* outperform BC-like methods; moreover, a vanilla combination of both actually *hurts* model performance. Under our framework, we study when BC is expected to help, and empirically validate our findings. Our work further bridges goal-reaching and generative modeling, illustrating the nuances and new pathways of extending the success of generative models to RL.

## 1 INTRODUCTION

Goal reaching is an essential aspect of intelligence in sequential decision making. Unlike the conventional formulation of reinforcement learning (RL), which aims to encode all desired behaviors into a single scalar reward function that is amenable to learning (Silver et al., 2021), goal reaching formulates the problem of RL as applying a sequence of actions to *rearrange* the environment into a desired state (Batra et al., 2020). Goal-reaching is a highly flexible formulation. For instance, we can design the goal-space to capture salient information about specific factors of variations that we care about (Plappert et al., 2018); we can use natural language instructions to define more abstract goals (Lynch & Sermanet, 2020; Ahn et al., 2022); we can encourage exploration by prioritizing previously unseen goals (Pong et al., 2019; Warde-Farley et al., 2018; Pitis et al., 2020); and we can even use self-supervised procedures to naturally learn goal-reaching policies without reward engineering (Pong et al., 2018; Nair et al., 2018b; Zhang et al., 2021; OpenAI et al., 2021).

Reward is *not* enough. Usually, rewards are manually constructed, either through laborious reward engineering, or from task-specific optimal demonstrations, neither of which is a scalable solution. How can RL agents learn useful behaviors from unlabeled reward-free trajectories, similar to how NLP models such as BERT (Devlin et al., 2018) and GPT-3 (Brown et al., 2020) are able to learn language from unlabeled text corpus? Goal reaching is a promising paradigm for unsupervised behavior acquisition, but it is unclear how to write down a well-defined objective for goal-conditioned policies, unlike language models that just predict the next token. This paper aims to define such an objective that unifies many prior approaches and strengthens the foundation of goal-conditioned RL.

We start from the following observation: hindsight goal relabeling (Andrychowicz et al., 2017) can turn an arbitrary trajectory into a sub-optimal expert demonstration. Thus, goal-conditioned RL might be doing a special kind of imitation learning. Currently, divergence minimization is the *de facto* way to describe imitation learning methods (Ghasemipour et al., 2020), so we should be able to recast hindsight goal relabeling into the divergence minimization framework. On top of that, to tackle the *sub-optimality* of hindsight-relabeled trajectories, we should explicitly maximize the probability of

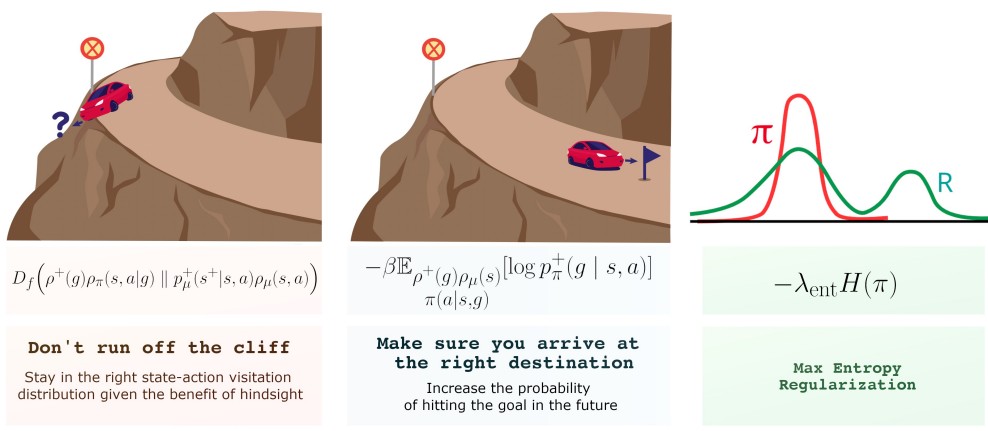

Figure 1: A unified objective for goal-reaching (see equation equation 14 for more details).

reaching the desired goal. Following these intuitions, we derive the reward function used in HER from first principles, as well as other behaviour cloning (BC) like methods such as GCSL (Ghosh et al., 2019) and Hindsight BC (HBC) (Ding et al., 2019).

Experimentally, we show that multi-goal $Q$-learning based on HER-like rewards, when carefully tuned, can *still* outperform goal-conditioned BC (such as GCSL / HBC). Moreover, a vanilla combination of multi-goal $Q$-learning and BC (HER + HBC), supposedly combining the best of both worlds, in fact *hurts* performance. We utilize our unified framework to analyze when a BC loss could help, and propose a modified algorithm named Hindsight Divergence Minimization (HDM) that uses $Q$-learning to account for the worst while imitating the best. HDM avoids the pitfalls of HER + HBC and improves policy success rates on a variety of self-supervised goal-reaching environments. Additionally, our framework reveals a largely unexplored design space for goal-reaching algorithms and potential paths of importing generative modeling techniques into multi-goal RL.

## 2 REINFORCEMENT LEARNING AND GENERATIVE MODELING

### 2.1 THE REINFORCEMENT LEARNING (RL) PROBLEM

We first review the basics of RL and generative modeling. A Markov Decision Process (MDP) is typically parameterized by $(\mathcal{S}, \mathcal{A}, \rho^0, p, r)$: a state space $\mathcal{S}$, an action space $\mathcal{A}$, an initial state distribution $\rho^0(s)$, a dynamics function $p(s' \mid s, a)$ which defines the transition probability, and a reward function $r(s, a)$. A policy function $\mu$ defines a probability distribution $\mu : \mathcal{S} \times \mathcal{A} \to \mathbb{R}^+$. For an infinite-horizon MDP, given the policy $\mu$, and the state distribution at step $t$ (starting from $\rho^0$ at $t = 0$), the state distribution at step $t + 1$ is given by:

$$\rho_\mu^{t+1}(s') = \int_{\mathcal{S} \times \mathcal{A}} p(s' \mid s, a) \mu(a \mid s) \rho_\mu^t(s) \mathrm{d}s \mathrm{d}a \tag{1}$$

The *state visitation distribution* sums over all timesteps via a geometric distribution $\mathrm{Geom}(\gamma)$:

$$\rho_\mu(s) = (1 - \gamma) \cdot \sum_{t=0}^{\infty} \gamma^t \cdot \rho_\mu^t(s) \tag{2}$$

However, the trajectory sampling process does not happen in this discounted manner, so the discount factor $\gamma \in (0, 1)$ is often absorbed into the cumulative return instead (Silver et al., 2014):

$$\mathcal{J}(\mu) = \frac{1}{1 - \gamma} \int_{\mathcal{S}} \rho_\mu(s) \int_{\mathcal{A}} \mu(a \mid s) r(s, a) \mathrm{d}a \mathrm{d}s = \mathop{\mathbb{E}}_{\substack{\rho^0(s_0)\mu(a_0|s_0) \\ p(s_1|s_0,a_0)\mu(a_1|s_1)\cdots}} \left[ \sum_{t=0}^{\infty} \gamma^t r(s_t, a_t) \right] \tag{3}$$

From 1 and 2, we also see that the future state distribution $p^+$ of policy $\mu$, defined as a geometrically discounted sum of state distribution at all future timesteps given current state and action, is given by

the following recursive relationship (Eysenbach et al., 2020b; Janner et al., 2020):

$$p_\mu^+(s^+ \mid s, a) = (1 - \gamma)p(s^+ \mid s, a) + \gamma \int_{S \times A} p(s' \mid s, a)\mu(a' \mid s')p_\mu^+(s^+ \mid s', a')\mathrm{d}s'\mathrm{d}a' \quad (4)$$

In multi-goal RL, an MDP is augmented with a goal space $\mathcal{G}$, and we learn a goal-conditioned policy $\pi : \mathcal{S} \times \mathcal{G} \times \mathcal{A} \to \mathbb{R}^+$. Hindsight Experience Replay (HER) (Andrychowicz et al., 2017) gives the agent a reward of 0 when the goal is reached and $-1$ otherwise, and uses *hindsight goal relabeling* to increase learning efficiency of goal-conditioned $Q$-learning by replacing the initial *behavioral goals* with *achieved goals* (future states within the same trajectory).

## 2.2 Imitation Learning (IL) as Divergence Minimization

We first review $f$-divergence between two probability distributions $P$ and $Q$ and its variational bound:

$$D_f(P \parallel Q) = \int_X q(x)f\left(\frac{p(x)}{q(x)}\right)\mathrm{d}x \geq \sup_{T \in \mathcal{T}} \mathbb{E}_{x \sim P}[T(x)] - \mathbb{E}_{x \sim Q}[f^*(T(x))] \quad (5)$$

where $f$ is a convex function such that $f(1) = 0$, $f^*$ is the convex conjugate of $f$, and $T$ is an arbitrary class of functions $T : X \to \mathbb{R}$. This variational bound was originally derived in (Nguyen et al., 2010) and was popularized by GAN (Goodfellow et al., 2014; Nowozin et al., 2016) and subsequently by imitation learning (Ho & Ermon, 2016; Fu et al., 2017; Finn et al., 2016; Ghasemipour et al., 2020). The equality holds true under mild conditions (Nguyen et al., 2010), and the optimal $T$ is given by $T^*(x) = f'(p(x)/q(x))$. The canonical formulation of imitation learning follows (Ho & Ermon, 2016; Ghasemipour et al., 2020), where $\rho^{\mathrm{exp}}(s, a)$ is from the expert:

$$\min_\mu D_f(\rho^{\mathrm{exp}}(s, a) \parallel \rho^\mu(s, a)) \Leftrightarrow \min_\mu \max_T \mathbb{E}_{\rho^{\mathrm{exp}}(s,a)}[T(s, a)] - \mathbb{E}_{\rho^\mu(s,a)}[f^*(T(s, a))] \quad (6)$$

Because of the *policy gradient theorem* (Sutton et al., 1999), the policy $\mu$ needs to optimize the *cumulative* return under its own trajectory distribution $\rho^\mu(s, a)$ with the reward being $r(s, a) = f^*(T(s, a))$. Under this formulation, Jensen-Shannon divergence leads to GAIL (Ho & Ermon, 2016), reverse KL leads to AIRL (Fu et al., 2017). Note that $f$ can in principle be any convex function (we can satisfy $f(1) = 0$ by simply adding a constant).

## 2.3 Energy-Based Models (EBM)

An energy-based model (EBM) is defined by $p_\theta(x) = \dfrac{\exp(-E_\theta(x))}{Z(\theta)}$, where $E_\theta : \mathbb{R}^D \to \mathbb{R}$ is the energy function and $Z(\theta) = \int_{\mathcal{X}} \exp(-E_\theta(x))\mathrm{d}x$ is the partition function. The gradient of the log-likelihood $\log p_\theta(x)$ w.r.t. $\theta$ (known as contrastive divergence (Hinton, 2002)) can be expressed as :

$$\frac{\partial \log p_\theta(x)}{\partial \theta} = \mathbb{E}_{p_\theta(x')}\left[\frac{\partial E_\theta(x')}{\partial \theta}\right] - \frac{\partial E_\theta(x)}{\partial \theta} \quad (7)$$

Sampling from $p_\theta(x)$ can be difficult, as Langevin dynamics is often required (Welling & Teh, 2011; Du & Mordatch, 2019; Grathwohl et al., 2019). Alternatively, EBMs can be trained via Noise Contrastive Estimation (NCE) (Gutmann & Hyvärinen, 2012; Mnih & Kavukcuoglu, 2013; Gao et al., 2020; Rhodes et al., 2020), which assumes access to a "noise" distribution $p_n(x)$ and learns energy functions through density ratio estimation. Let $s_\theta(x) = -E_\theta(x) - \log Z_\theta$ (Mnih & Teh, 2012). NCE learns a classifier that distinguishes the data distribution $p(x)$ from noise $p_n(x)$, where noise samples are $k$ times more frequent (Mnih & Kavukcuoglu, 2013; Mikolov et al., 2013):

$$p(D = 1 \mid x) = p_\theta(x)/(p_\theta(x) + k \cdot p_n(x)) = \sigma(s_\theta(x) - \log p_n(x) - \log k) \quad (8)$$

To be more concise, we denote $\Delta_\theta(x, k) = s_\theta(x) - \log p_n(x) - \log k$. Gradients of the weighted binary cross entropy loss asymptotically approximate contrastive divergence (7) (Mnih & Teh, 2012):

$$\frac{d}{d\theta}\left[\mathbb{E}_{p(x)}[\log \sigma(\Delta_\theta(x, k))] + k \cdot \mathbb{E}_{p_n(x)}[\log(1 - \sigma(\Delta_\theta(x, k)))]\right] \xrightarrow{k \to \infty} \frac{\partial \log p_\theta(x)}{\partial \theta} \quad (9)$$

Closely related to NCE is the more recent InfoNCE loss (Van den Oord et al., 2018) which has gained popularity in the contrastive learning setting (Chen et al., 2020; He et al., 2020); we can interpret $k$ in NCE as the batch size of *negative samples* in InfoNCE. In summary, NCE gives EBM a more tractable way to maximize data likelihood.

## 3 GRAPHICAL MODELS FOR HINDSIGHT GOAL RELABELING

Consider the setting: given an environment $\xi = (\rho^0(s), p(s' \mid s, a), \rho^+(g))$ where we generate a dataset of trajectories $\mathcal{D} = \{(s_0, a_0, s_1, a_1, \cdots)\}$ by sampling from the initial state distribution $\rho_0(s)$, an unobserved actor policy $\mu(a \mid s)$, and the dynamics $p(s' \mid s, a)$. We aim to train a goal-conditioned policy $\pi(a \mid s, g)$ from this arbitrary dataset with relabeled future states as the goals. $\rho^+(g)$ is the behavioral goal distribution assumed to be given *apriori* by the environment. To recast the problem of goal-reaching as imitation learning equation 6, we need to set up an $f$-divergence minimization where we define the target (expert) distribution and the policy distribution we want to match.

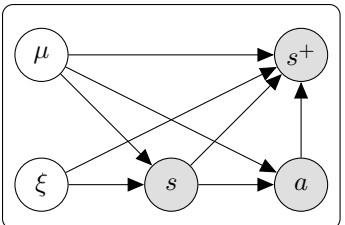 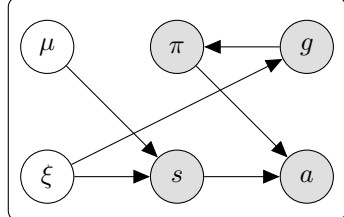 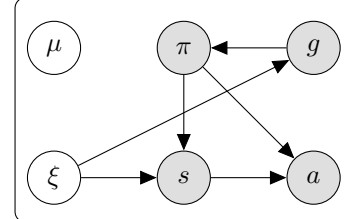

Figure 2: $\rho_\mu(s, a)p_\mu^+(s^+ \mid s, a)$ hindsight-relabeled distribution    Figure 3: $\rho^+(g)\rho_\mu(s)\pi(a \mid s, g)$ (used in HBC / GCSL)    Figure 4: $\rho^+(g)\rho_\pi(s, a \mid g)$ (used in HER / HDM)

The training signal comes from factorizing the joint distribution of state-action-goal differently. For the relabeled target distribution, we assume an unconditioned actor $\mu$ generating a state-action distribution at first, with the goals coming from the future state distribution equation 4 conditioned on the given state and action. For the goal-conditioned policy distribution, behavior goals are given *apriori*, and the state-action distribution is generated conditioned on the behavioral goals. Thus, the *target distribution* (see Figure 2) for states, actions, and *hindsight goals* is:

$$p_\mu(s, a, s^+) = \rho_\mu(s, a)p_\mu^+(s^+ \mid s, a) \tag{10}$$

Note that $p_\mu^+$ is given by equation equation 4, and $\rho_\mu(s, a)$ is similar to $\rho^{\exp}(s, a)$ in equation equation 6. In the fashion of behavioral cloning (BC), if we do not care about matching the states, we can write the joint distribution we are trying to match as (see Figure 3):

$$p_\pi^{\text{BC}}(s, a, g) = \rho^+(g)\rho_\mu(s)\pi(a \mid s, g) \tag{11}$$

We can recover the objective of Hindsight Behavior Cloning (HBC) (Ding et al., 2019; Eysenbach et al., 2020a) and Goal-Conditioned Supervised Learning (GCSL) Ghosh et al. (2019) via minimizing a KL-divergence:

$$\min_\pi \mathcal{D}_{KL}\Big(p_\mu(s, a, s^+) \parallel p_\pi^{\text{BC}}(s, a, g)\Big) \Leftrightarrow \min_\pi \mathbb{E}_{\rho_\mu(s,a)p_\mu^+(s^+\mid s,a)}[-\log \pi(a \mid s, g)] \tag{12}$$

In many cases, matching the states is more important than matching state-conditioned actions (Ross et al., 2011; Ghasemipour et al., 2020). The joint distribution for states, actions, *behavioral goals* for $\pi$ (see Figure 4) is:

$$p_\pi(s, a, g) = \rho^+(g)\rho_\pi(s, a \mid g) \tag{13}$$

However, it is important to recognize that even after adding this state-matching part, there is still a missing component of the objective. Divergence minimization encourages the agent to stay in the "right" state-action distribution given the benefit of hindsight, but we still need the policy to actually hit the goal (see Figure 1). When we condition the policy on a goal, we should maximize the likelihood of seeing that goal in the future state distribution (see Figure 5). Together with a maximum entropy regularization on the policy $\mathcal{H}(\pi)$ (Ho & Ermon, 2016; Schulman et al., 2017), we propose the following goal-reaching objective:

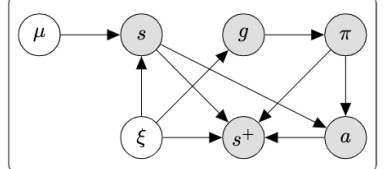

Figure 5: the future state distribution $p_\pi^+(s^+ \mid s, a)$ of a goal-conditioned policy when starting from a state $s$ visited by $\mu$.

$$\min_\pi \underbrace{D_f\Big(\rho^+(g)\rho_\pi(s, a|g) \parallel p_\mu^+(s^+|s, a)\rho_\mu(s, a)\Big)}_{(a) \; f\text{-divergence term}} - \underbrace{\beta\mathbb{E}_{\substack{\rho^+(g)\\ \rho_\mu(s)\\ \pi(a|s,g)}}\left[\log p_\pi^+(g|s, a)\right]}_{(b) \text{ goal likelihood term}} - \underbrace{\lambda \; \mathcal{H}(\pi)}_{(c) \text{ entropy}} \tag{14}$$

Compared to equation 12 and equation 6, the $f$-divergence term equation 14(a) swaps the order between the expert and the policy. The coefficient $\beta$ controls the importance of "hitting the goal". This incentive is already implicit in hindsight-relabeled data, as in many cases a BC objective alone can learn to hit the goal as well (Ding et al., 2019; Ghosh et al., 2019; Lynch et al., 2019; Jang et al., 2021). However, hindsight-relabeled trajectories are often *sub-optimal demonstrations*: not every action is acting towards achieving the goal. Thus, maximizing the likelihood of achieving the goal is crucial, and it plays a key role in deriving the reward function of HER (Andrychowicz et al., 2017).

## 4 BRIDGING GOAL-REACHING AND GENERATIVE MODELING

In this section, we study how to optimize the proposed unifying objective 14. We show that the $Q$-function trained for goal-reaching $Q_\theta(s, a, g)$ is implicitly doing generative modeling. Its temporal difference is modeling the density ratio between the hindsight-relabeled distribution and the goal-conditioned behavioral distribution. It also approximates an energy-based model for future-state distribution when marginalized over the goal distribution. In section 4.1, we demonstrate how goal-conditioned $Q$-learning does divergence minimization (equation 14(a)). In section 4.2, we show how $Q$-functions can define an EBM for cumulative future states, which in turn allows the policy to maximize goal likelihood (equation 14 b+c). In section 4.3, we show that combining the derived results yields the HER reward. In section 4.4, we study when a BC loss can help by analyzing the loss terms being left out by the HER reward.

### 4.1 DIVERGENCE MINIMIZATION WITH GOAL-CONDITIONED $Q$-LEARNING

This section decomposes equation 14(a). We start with the $f$-divergence bound from equations equation 5 and equation 6.

$$D_f(p_\pi(s, a, g) \parallel p_\mu(s, a, s^+)) = \max_T \mathbb{E}_{\substack{p(g) \\ \rho_\pi(s,a|g)}} [T(s, a, g)] - \mathbb{E}_{\substack{\rho_\mu(s,a) \\ p_\mu^+(s^+|s,a)}} [f^*(T(s, a, s^+))] \quad (15)$$

Now we negate $T$ to get $r(s, a, g) = -T(s, a, g)$, and the divergence minimization problem becomes:

$$\max_\pi \min_r \mathbb{E}_{\substack{\rho_\mu(s,a) \\ p_\mu^+(s^+|s,a)}} [f^*(-r(s, a, s^+))] + \mathbb{E}_{\substack{p(g) \\ \rho_\pi(s,a|g)}} [r(s, a, g)] \quad (16)$$

We can interpret $r$ as a GAIL-style (Ho & Ermon, 2016) discriminator or reward. However, we aim to derive a discriminator-free learning process that directly trains the $Q$-function corresponding to this reward $Q_\theta(s, a, g) = r(s, a, g) + \gamma \cdot \mathcal{P}^\pi Q(s, a, g)$ where $\mathcal{P}^\pi$ is the *transition operator*: $\mathcal{P}^\pi Q(s, a, g) = \mathbb{E}_{p(s'|s,a)\pi(a'|s',g)}[Q_\theta(s', a', g)]$. Re-writing the equation equation 16 w.r.t $Q$:

$$\min_Q \mathbb{E}_{\substack{\rho_\mu(s,a) \\ p_\mu^+(s^+|s,a)}} [f^*(-(Q_\theta - \gamma \cdot \mathcal{P}^\pi Q)(s, a, s^+))] + \mathbb{E}_{\substack{p(g) \\ \rho_\pi(s,a|g)}} [(Q_\theta - \gamma \cdot \mathcal{P}^\pi Q)(s, a, g)] \quad (17)$$

A similar change-of-variable has been explored in the context of offline RL (Nachum et al., 2019a;b) and imitation learning (Kostrikov et al., 2019; Zhu et al., 2020); we may call those methods the DICE (Nachum & Dai, 2020) family. The major pain point of DICE-like methods is that they require samples from the initial state distribution $\rho^0(s)$ (Garg et al., 2021). Here, the following lemma shows that in the goal-conditioned case, we can use arbitrary offline trajectories to evaluate the expected rewards under goal-conditioned online rollouts:

**Lemma 4.1** (Online-to-offline transformation for goal reaching). *Given a goal-conditioned policy $\pi(a \mid s, g)$, its corresponding $Q$-function $Q^\pi(s, a, g)$, and arbitrary state-action visitation distribution $\rho_\mu(s, a)$ of another policy $\mu(a \mid s)$, the expected temporal difference for online rollouts under $\pi$ is:*

$$\mathbb{E}_{p(g)\rho_\pi(s,a|g)}[(Q^\pi - \gamma \cdot \mathcal{P}^\pi Q^\pi)(s, a, g)] = \mathbb{E}_{p(g)\rho_\mu(s,a)\pi(\tilde{a}|s,g)}[Q^\pi(s, \tilde{a}, g) - \gamma \cdot \mathcal{P}^\pi Q^\pi(s, a, g)]$$

Using Lemma 4.1, the objective in equation 16 now becomes:

$$\max_\pi \min_Q \mathbb{E}_{\substack{\rho_\mu(s,a)p_\mu^+(s^+|s,a) \\ p(g), \pi(\tilde{a}|s,g)}} \left[ f^*(-(Q_\theta - \gamma \mathcal{P}^\pi Q)(s, a, s^+)) + Q_\theta(s, \tilde{a}, g) - \gamma \mathcal{P}^\pi Q(s, a, g) \right] \quad (18)$$

The function $f^*$ is the convex conjugate of $f$ in $f$-divergence. We can pick almost any convex function as $f^*$ as long as $((f^*)^*)'(1) = 0$. For instance, in the tradition of $Q$-learning, we can use:

$$f^*(x) = (x + \bar{r})^2/2 + \bar{c} \quad (19)$$

where $\overline{r}$ and $\overline{c}$ are constants. Its convex conjugate is $f(x) = (f^*)^*(x) = x^2/2 - \overline{r}x - c$. For any $\overline{r}$, we can always pick a $\overline{c}$ that will ensure $f(1) = (f^*)^*(1) = 0$, and $\overline{c}$ does not affect the learning process. In summary, we have derived a way to minimize the first term $\mathcal{D}_f(p_\mu(s, a, s^+) \parallel p_\pi(s, a, g))$ in equation 14 directly using goal-conditioned $Q$-learning.

## 4.2 EBM FOR PREDICTING (AND CONTROLLING) THE FUTURE

In this section, we study how to maximize the goal likelihood term equation 14(b) by defining an energy-based model (EBM) to *predict* the future-state distribution for a policy, which the policy can in turn optimize to *control* the future-state distribution. We start with the Bayes rule:

$$p_\pi^+(s^+ \mid s, a) = \frac{\rho^+(s^+)\rho_\pi(s, a \mid s^+)}{\mathbb{E}_{\rho^+(g)}[\rho_\pi(s, a \mid g)]} \tag{20}$$

This relationship reflects that, if we define an EBM for $\rho_\pi(s, a \mid s^+) = \exp q(s, a, s^+)/Z_q(s^+)$ where $Z_q(s^+) = \int_{\mathcal{S} \times \mathcal{A}} \exp q(s, a, s^+)\mathrm{d}s\mathrm{d}a$, then we can contrast the dynamics defined in equation 4 with the marginal goal distribution defined in equation 13. In a slightly overloaded notation, we will now define a $Q$-function as $Q_\theta(s, a, g) = q(s, a, g) - \log Z_q(g)$. For now, we shall assume that this $Q$-function is separate from the one defined in equation 18. Rearranging equation 20 , and setting $\rho^+(g)$ to be $\rho^+(s^+)$, we see the density ratio can be expressed as:

$$\frac{p_\pi^+(s^+ \mid s, a)}{\rho^+(s^+)} = \frac{\rho_\pi(s, a \mid s^+)}{\mathbb{E}_{\rho^+(g)}[\rho_\pi(s, a \mid g)]} = \frac{\exp Q_\theta(s, a, s^+)}{\mathbb{E}_{\rho^+(g)}[\exp Q_\theta(s, a, g)]} \tag{21}$$

**Lemma 4.2** (Gradient of the noise-contrastive term in energy-based goal-reaching)**.** *Given the following definition for the logit of a NCE-like binary classifier, with $\rho^+(\overline{g}) = \rho^+(g)$:*

$$\Delta_\theta(s, a, g, k) = Q_\theta(s, a, g) - \log \mathbb{E}_{\rho^+(\overline{g})}[\exp Q_\theta(s, a, \overline{g})] - \log k \tag{22}$$

*The gradient of the negative NCE term in the density ratio estimation approaches zero as $k \to \infty$:*

$$\frac{d}{d\theta}\mathbb{E}_{\rho_\mu(s,a)\rho^+(g)}\Big[k \cdot \log\Big(1 - \sigma(\Delta_\theta(s, a, g, k))\Big)\Big] \xrightarrow{k \to \infty} 0$$

As for the positive classification term in NCE equation 9, we can make a similar argument that $\nabla_\theta \log(1 + \exp\Delta_\theta(s, a, s^+, k)) \xrightarrow{k \to \infty} 0$ (see appendix B.2). We have:

$$\arg\max_Q \mathbb{E}_{\rho_\mu(s,a)}\Big[\mathbb{E}_{p_\pi^+(s^+\mid s,a)}[Q_\theta(s, a, s^+)] - \log\mathbb{E}_{\rho^+(g)}[\exp Q_\theta(s, a, g)]\Big] \tag{23}$$

To optimize the above objective on arbitrary behaviour data, we have to consider the fact that we do not have complete access to the distribution of "positive samples" $p_\pi^+(s^+ \mid s, a)$, as sampling directly from this distribution requires on-policy rollouts. Utilizing importance weights to learn from off-policy data while accounting for hindsight bias yields the following loss minimization instead:

$$\mathbb{E}_{\rho_\mu(s,a)}\Big\{ \underbrace{-(1-\gamma) \cdot \mathbb{E}_{p(s'\mid s,a)}[Q_\theta(s, a, s')]}_{\text{Learning single-step dynamics in } Q} + \underbrace{\mathbb{E}_{\rho^+(g)p(s'\mid s,a)}[w(s, a, s', g) \cdot Q_\theta(s, a, g)]}_{\text{Learning multi-step dynamics in } Q} \Big\} \tag{24}$$

where $w(s, a, s', g)$ contrasts the goals that will likely be met by the agent *beyond* a single step of dynamics $p(s' \mid s, a)$ in the future with other random goals that the agent has seen in the past:

$$w(s, a, s', g) = \underbrace{\frac{\exp Q(s, a, g)}{\mathbb{E}_{\rho^+(\overline{g})}[\exp Q(s, a, \overline{g})]}}_{\text{Pushing down the likelihood of the marginal}} - \gamma \cdot \underbrace{\frac{\sum_{\overline{a}} \exp Q(s', \overline{a}, g)/|\mathcal{A}|}{\mathbb{E}_{\rho^+(\overline{g})}[\sum_{\overline{a}} \exp Q(s', \overline{a}, \overline{g})/|\mathcal{A}|]}}_{\text{Pushing up the likelihood of the conditional}} \tag{25}$$

See Appendix B.3 for a full derivation.

## 4.3 DERIVING HER REWARDS

For a goal-conditioned $Q$-function $Q_\theta(s, a, g)$, besides optimizing a Bellman residual as a regular $Q(s, a)$ function would do, it has an additional degree-of-freedom for the goal $g$ which can be used to

define an EBM that models the future-state distribution. In such an EBM, the $Q$-function is used as the negative energy defined in equation 21. Combining the first term in the $f$-divergence minimization part equation 18 and the single-step dynamics loss term in the EBM learning 24 gives us:

$$\arg\min_{Q} \mathbb{E}_{\rho_\mu(s,a)p(s'|s,a)p_\mu^+(s^+|s,a)}\left[f^*(-(Q_\theta - \gamma\mathcal{P}^\pi Q)(s,a,s^+)) - \beta \cdot (1-\gamma)Q_\theta(s,a,s')\right] \quad (26)$$

The crucial observation here is, given that $f^*$ is a quadratic 19, and that there is a stop gradient sign on $\mathcal{P}^\pi Q$ because it uses a target network, the loss above can be re-packaged into a single quadratic loss of Bellman residuals for a specific reward thanks to the property of the dynamics defined in 4:

$$\arg\min_{Q} \mathbb{E}_{\rho_\mu(s,a)p(s'|s,a)p_\mu^+(s^+|s,a)}\left[\frac{1}{2}\Big(r(s,a,s',s^+) + (\gamma\mathcal{P}^\pi Q - Q_\theta)(s,a,s^+)\Big)^2\right] \quad (27)$$

$$r(s,a,s',s^+) = \begin{cases} \overline{r} + \beta, & s' = s^+ \\ \overline{r}, & s' \neq s^+ \end{cases} \quad (28)$$

See Appendix B.4 for a full derivation. Setting $\overline{r} = -1$ and $\beta = 1$, we have arrived at the reward function of HER $r_{\text{HER}}(s,a,s',s^+)$. The remaining losses that this reward leaves out are the second term in $f$-divergence minimization part equation 18 and the second term in the EBM training equation 53, which we will analyze in the next section.

## 4.4 WHEN DOES BEHAVIOUR CLONING (BC) HELP?

We now proceed to answer the following question: when does BC help? Our idea is to identify the conditions for a BC-like loss to emerge in the remaining losses unaccounted by the HER reward: a loss term in 18 about pushing $Q$-values of the current policy down, and the multi-step dynamics term in EBM 54. We may argue that, by not pushing the $Q$-values of the current policy down, the HER agent becomes more exploratory; by not pushing up the $Q$-values of the discounted future-state distribution beyond a single step, the HER agent is encouraged to reach a goal sooner rather than later. Nevertheless, we shall see that combining those two remaining terms produces a BC-like term, which *imitates the best actions* based on how much an action moves the agent closer to the goal.

We start with the following property of Boltzmann policies $\pi(a \mid s,g) \propto \exp Q(s,a,g)$ (Haarnoja et al., 2017; Schulman et al., 2017) (which we can apply thanks to entropy regularization in 14):

$$\mathbb{E}_{\pi(a|s,g)}[Q(s,a,g)] = \log\sum_{a}\exp Q(s,a,g) - \mathcal{H}(\pi) \quad (29)$$

In equation 25, the $Q$-value of a particular action $a$ is *pushed up* under the following condition:

$$\frac{\exp Q(s,a,g)}{\mathbb{E}_{\rho^+(\overline{g})}[\exp Q(s,a,\overline{g})]} < \gamma \cdot \frac{\sum_{\overline{a}}\exp Q(s',\overline{a},g)/|\mathcal{A}|}{\mathbb{E}_{\rho^+(\overline{g})}[\sum_{\overline{a}}\exp Q(s',\overline{a},\overline{g})/|\mathcal{A}|]} \quad (30)$$

Both denominators on the left and right sides are averaged over all possible behavioral goals, so their values should be roughly the same. Moreover, we approximate the average-exponential operation by using the max operation already computed in the backup operator (either exact (Mnih et al., 2015; Van Hasselt et al., 2016) or approximate (Lillicrap et al., 2015; Haarnoja et al., 2018)) and lower the threshold $\gamma_{\text{hdm}}$ (compared to $\gamma$) accordingly. After those changes, the indicator function that decides whether the value of an action $a$ should be pushed up becomes:

$$\hat{w}(s,a,s',g) = \mathbb{1}(\exp Q(s,a,g) - \gamma_{\text{hdm}} \cdot \exp\max_{a'} Q(s',a',g) < 0) \quad (31)$$

Combining with the first term in 29, we arrive at a BC-like loss 12, but with an indicator weighting:

$$\mathcal{L}_{\text{hdm}}(Q) = \mathbb{E}_{\rho_\mu(s,a)p(s'|s,a)\rho^+(g)}[-\hat{w}(s,a,s',g) \cdot (Q_\theta(s,a,g) - \log\sum_{\mathcal{A}}\exp Q_\theta(s,\cdot,g))] \quad (32)$$

Intuitively, this term imitates a particular action when the value functions believe that this action can move the agent closer to the goal by at least $-\log\gamma_{\text{hdm}}$ steps. The idea of *imitating the best actions* is similar to self-imitation learning (SIL) (Oh et al., 2018; Vinyals et al., 2019), but our algorithm 1 operates in a (reward-free) goal-reaching setting, with the advantage function in SIL being replaced by the *delta* of reachability a particular action can produce in getting closer to the goal.

---

**Algorithm 1** Hindsight Divergence Minimization

---

**Given**: Batch of data $\{(s, a, s', s^+)\}$, where $s^+$ is sampled via hindsight relabeling.

1: $\mathcal{L} = \texttt{MSE}(Q(s, a, s^+) - \gamma \max_{a'} \hat{Q}(s', a', s^+), \bar{r})$      $\triangleright \bar{r}$ is a constant, default $\bar{r} = -1$
2: $\mathcal{L} = \mathcal{L} - \beta(1 - \gamma_{\text{hdm}}) \cdot Q(s, a, s')$      $\triangleright$ Push up the values for reaching next states
3: $\mathcal{L}_{BC} = Q(s, a, s^+) - \log \sum_{\mathcal{A}} \exp Q(s, \cdot, s^+)$      $\triangleright$ Behaviour Cloning like loss
4: $\mathcal{L} = \mathcal{L} - \beta \cdot \mathbb{1}(Q(s, a, s^+) - \max_{a'} Q(s', a', s^+) < \log \gamma_{\text{hdm}}) \cdot \mathcal{L}_{BC}$
5: **minimize** $\mathcal{L}$, update the target network $\hat{Q}$

---

## 5 EXPERIMENTS

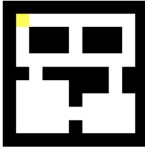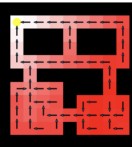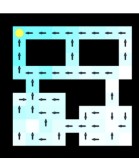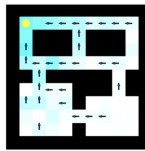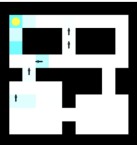

Figure 6: Intuitions about when HDM applies BC. From left to right: (a) the Maze environment with a goal in the upper-left corner; (b) the $Q$-values learned through the converged policy. Lighter red means higher Q-value; (c) visualizing the actions (from the replay) that get imitated when conditioning on the goal and setting $\gamma_{\text{hdm}} = 0.95$, with the background color reflecting how much an action moves the agent closer to the goal *based on the agent's own estimate*; (d) $\gamma_{\text{hdm}} = 0.85$; (e) $\gamma_{\text{hdm}} = 0.75$. As we lower $\gamma_{\text{hdm}}$, the threshold $-\log \gamma_{\text{hdm}}$ gets higher and fewer actions get imitated, with the remaining imitated actions more concentrated around the goal. HDM uses $Q$-learning to account for the worst while imitating the best during the goal-reaching process.

### 5.1 SELF-SUPERVISED GOAL-REACHING SETUP

We consider the self-supervised goal-reaching setting for the environments described in GCSL (Ghosh et al., 2019). While the original HER (Andrychowicz et al., 2017) assumes that the agent has direct access to the ground truth binary reward metric (which is used when relabeling is performed), we do not make such an assumption as it can be unrealistic for real-world robot-learning (Lin et al., 2019). Instead, we simply use *next-state relabeling* to provide positive rewards. As seen in Eq equation 28, a positive reward is provided only when the relabeled hindsight goal is the immediate next state. The benefit of the *next-state relabeling* reward is that the training procedure is now completely self-supervised, and therefore provides a fair comparison to GCSL (Ghosh et al., 2019).

| Success Rate (%) | Four Rooms | Lunar Landar | Sawyer Push | Door Opening | Claw Manipulate |
|---|---|---|---|---|---|
| GCSL / HBC | 78.27 ±4.76 | 50.00 ±7.77 | 44.67 ±13.86 | 19.10 ±5.97 | 16.80 ±6.55 |
| HER $r = (0, 1)$ | 86.60 ±4.22 | 39.30 ±6.81 | 57.60 ±6.61 | 82.50 ±4.87 | 22.80 ±6.43 |
| HER + SQL | 88.50 ±4.56 | 44.50 ±9.61 | 57.20 ±6.32 | 84.70 ±5.33 | 16.13 ±8.43 |
| HER $r = (-1, 0)$ | 86.40 ±5.11 | 50.80 ±4.66 | 54.60 ±6.16 | 83.76 ±6.02 | 20.20 ±6.23 |
| HER + HBC | 82.90 ±6.24 | 35.33 ±4.57 | 52.63 ±8.05 | 76.44 ±5.37 | 16.93 ±8.03 |
| HDM (ours) | 96.27 ±2.56 | 57.60 ±7.21 | 66.00 ±5.13 | 88.60 ±4.63 | 27.89 ±6.46 |

Table 1: Benchmark results of test-time success rates in self-supervised goal-reaching. We compare our method HDM with GCSL (Ghosh et al., 2019), HER (Andrychowicz et al., 2017) with two different types of rewards, SQL (Schulman et al., 2017) (Soft Q-Learning) + HER, and HER + HBC. We find that HER can still outperforms GCSL / HBC, and a vanilla combination of HER + HBC actually hurts performance. HDM selectively decides on what to imitate and outperforms HER and HBC.

### 5.2 COMPARISONS AND ABLATIONS

As shown in Table 1, we compare HDM to the following baselines in terms of their goal-reaching abilities: GCSL (Ghosh et al., 2019) / HBC (Ding et al., 2019), HER (Andrychowicz et al., 2017) with $(0, 1)$ rewards, HER with Soft Q-Learning (SQL) (Schulman et al., 2017), HER with $(-1, 0)$ rewards, and HER + HBC. HER with $(0, 1)$ reward gives the agent a reward of 1 when the goal is reached and 0 otherwise; HER with $(-1, 0)$ reward gives a reward of 0 when a goal is reached and $-1$ otherwise. Those two types of rewards lead to different learning dynamics because the $Q$-function is initialized to output values around 0. We also compare against HER + SQL, because soft $Q$-learning

Normalized Performance Gain over GCSL

Figure 7: Normalized performance gain over GCSL, calculated by normalizing the final performance difference between an algorithm and GCSL. We see that only HER with $(-1, 0)$ rewards and HDM consistently outperform GCSL, while HER + HBC performs worse than HER and sometimes GCSL as well.

is known to improve the robustness of learning (Haarnoja et al., 2018). HDM builds on top of HER with $(-1, 0)$ rewards and adds a BC-like loss with a clipping condition equation 32 such that only the actions that move an agent closer to the goal get imitated (see Figure 6). Indeed, combining HER with HBC (which blindly imitates all actions in hindsight) produces worse results than not imitating at all and only resorting to value learning; HDM allows for better control over what to imitate.

The results in Table 1 and Figure 7 show that HDM achieves the strongest performance on all environments, while reducing variances in success rates. Interestingly, there is no consensus best baseline algorithm, with all five algorithms achieving good results in some environments and subpar performance in others. In Figure 10 of the appendix, we ablate the additional hyper-parameters introduced by HDM: $\gamma_{\text{hdm}}$ in equation 31 and the $\beta$ term in equation 14. Intuitively, $\gamma_{\text{hdm}}$ controls the threshold that determines when an action is considered good enough for imitation, and $\beta$ controls the trade-off between minimizing $f$-divergence and maximizing future goal likelihood in equation 14. The ablation shows that HDM outperforms HER and GCSL across a variety of $\gamma_{\text{hdm}}$ and $\beta$ values.

## 5.3 WHY DOES HINDSIGHT BC SOMETIMES FAIL?

In this section, we aim to better understand whether the (under-whelming) performance of hindsight BC had anything to do with our specific setting. We report an interesting metric that correlates with the performance of GCSL / HBC, which can be measured prior to training a policy: *initial ag (achieved-goal) change ratio*. We define the *ag change ratio* of $\pi$ to be: the percentage of trajectories where the achieved goals in initial states $s_0$ are different from the achieved goals in final states $s_T$ under $\pi$. Most training signals are only created from *ag* changes, because they provide examples of how to *rearrange* an environment. We then define *initial ag change ratio* to be the ag change ratio of a random-acting policy $\pi_0$. Using notation from 28, it can be computed as $\mathbb{E}_{s_0 \cdots s_T \sim \pi_0}[-r_{\text{HER}}(\cdot, \cdot, s_0, s_T)]$.

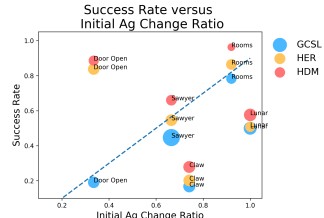

Figure 8: Success rate versus *initial achieved-goal* change ratio.

In Figure 8, we show the relationship between final success rates versus the initial ag (achieved-goal) change ratio, across all environments. The performance of GCSL seems to be upper-bounded by a linear relationship between the two. This makes intuitive sense: a BC-style objective starts off cloning the initially random trajectories, so if *initial ag change ratio* is low, the policy would not learn to rearrange *ag*, compounding to a low final performance. HER and HDM are able to surpass this upper ceiling likely because of the additional goal-likelihood term equation 14 besides imitation.

This finding suggests that in order to make goal-reaching *easier*, we should either modify the initial state distribution $\rho^0(s)$ such that *ag* can be easily changed through random exploration (Florensa et al., 2017) (if the policy is training from scratch), or initialize BC from some high-quality demonstrations where *ag* does change (Ding et al., 2019; Nair et al., 2018a; Lynch et al., 2019).

## 6 CONCLUSION

This work presents a unified goal-reaching objective that encompasses a family of goal-conditioned RL algorithms. Our derivation illustrates the connection between hindsight goal relabeling, divergence minimization, and energy-based models. It reveals that there is a largely unexplored design space: we could potentially use other convex functions in $f$-divergence minimization (Nowozin et al., 2016; Ghasemipour et al., 2020), and improve the optimization of the energy-based model (Du et al., 2020) by going beyond NCE (Grathwohl et al., 2020). The primary limitation of our framework is that it does not account for exploration (Hafner et al., 2020). Eventually, we would like to incorporate empowerment (Klyubin et al., 2005; Sekar et al., 2020) and input density exploration (Bellemare et al., 2016; Pong et al., 2019) into the framework.

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

## A    NOTATIONS

| | |
|---|---|
| $\rho^0$ | Initial state distribution |
| $p(s' \mid s, a)$ | Environmental dynamics |
| $p_\mu^+(s^+ \mid s, a)$ | Discounted future state distribution under policy $\mu$ |
| $\rho_\mu(s)$ | State distribution (occupancy measure) visited by policy $\mu$ |
| $\mathcal{P}^\pi Q(s, a, g)$ | Transition operator in (goal-conditioned) Bellman backup. $\mathcal{P}^\pi Q(s, a, g) = \mathbb{E}_{p(s'\mid s,a)\pi(a'\mid s',g)}[Q_\theta(s', a', g)]$. |
| $f$ | Convex function in $f$-divergence $D_f$. Usually $f(1) = 0$. |
| $f^*$ | Convex conjugate of function $f$ |
| $f'(x)$ | Derivative of function $f$ |
| $\rho^{\text{exp}}(s, a)$ | State-action visitation distribution of the expert policy |
| $\rho^\mu(s, a)$ | State-action visitation distribution of the policy $\mu$. Note that $\rho^\mu(s, a) = \rho^\mu(s)\mu(a \mid s)$. |
| $T(x)$ | Function used in the variational bound of $f$-divergence. Also appears as $T(s, a)$ or $T(s, a, g)$ in the main text. |
| $\rho^+(g)$ | The *behaviour* goal distribution assumed to be given *apriori* by the environment. |
| $\rho^+(s^+)$ | The marginal *hindsight* goal distribution of a given dataset / replay, where $\rho^+(s^+) = \mathbb{E}_{\rho_\mu(s,a)}[p_\mu^+(s^+ \mid s, a)]$ |
| $\hat{\pi}(a \mid s)$ | The goal-conditioned policy $\pi$ marginalized over the behavioral goal distribution $\hat{\pi}(a \mid s) = \mathbb{E}_{p^+(g)}[\pi(a \mid s, g)]$. |
| $\rho_\pi(s, a \mid g)$ | The state-action visitation distribution of goal-conditioned policy $\pi$ when conditioned on the behavioral goal $g$ |
| $E(x)$ | Energy function of an EBM: $p_\theta(x) = \exp(-E_\theta(x))/Z(\theta)$. |
| $Z(\theta)$ | The partition function in an EBM $Z(\theta) = \int_{\mathcal{X}} \exp(-E_\theta(x))\mathrm{d}x$. |
| $\sigma(x)$ | Sigmoid function $\sigma(x) = 1/(1 + \exp(-x))$. |
| $p_n(x)$ | Noise distribution in Noise Contrastive Estimation (NCE) |
| $k$ | The number of times noise samples are sampled more frequently than true data samples in NCE. |
| $\Delta_\theta$ | The logit of the positive sample classification loss in NCE. |
| $q(s, a, s^+)$ | A conditional EBM $\rho_\pi(s, a \mid s^+) = \exp q(s, a, s^+)/Z_q(s^+)$. |
| $Z_q(s^+)$ | The partition function of a conditional EBM $\rho_\pi(s, a \mid s^+)$: $Z_q(s^+) = \int_{\mathcal{S} \times \mathcal{A}} \exp q(s, a, s^+)\mathrm{d}s\mathrm{d}a$. |
| $\mathcal{H}(\pi)$ | Entropy of a policy across (replay) states and goals $\mathcal{H}(\pi) = \mathbb{E}_{\rho(s)\rho^+(g)\pi(a\mid s,g)}[-\log \pi(a \mid s, g)]$. |
| $r(s, a)$ | The (learned) reward in traditional RL settings without goal-conditioning. |
| $r(s, a, s^+)$ | The (learned) reward in goal-conditioned divergence minimization. |
| $r(s, a, s', s^+)$ | The (generalized) reward used in HER-style multi-goal RL. |
| $r_{\text{HER}}(s, a, s', s^+)$ | The reward function used in HER. It equals 0 when $s' = s^+$ and $-1$ when $s' \neq s^+$. Also denoted as $r_{\text{HER}}(\cdot, \cdot, s', s^+)$. |

# B PROOFS

## B.1 MAIN LEMMAS

**Lemma B.1** (Online-to-offline transformation for goal reaching). *Given a goal-conditioned policy* $\pi(a \mid s, g)$*, its corresponding Q-function* $Q^\pi(s, a, g)$*, and arbitrary state-action visitation distribution* $\rho_\mu(s, a)$ *of another policy* $\mu(a \mid s)$*, the expected temporal difference for online rollouts under* $\pi$ *is:*

$$\mathbb{E}_{p(g)\rho_\pi(s,a|g)}[(Q^\pi - \gamma \cdot \mathcal{P}^\pi Q^\pi)(s,a,g)] = \mathbb{E}_{p(g)\rho_\mu(s,a)\pi(\tilde{a}|s,g)}[Q^\pi(s,\tilde{a},g) - \gamma \cdot \mathcal{P}^\pi Q^\pi(s,a,g)]$$

*Proof of Lemma 4.1.*

$$\mathbb{E}_{p(g)\rho_\pi(s,a|g)}[(Q^\pi - \gamma \cdot \mathcal{P}^\pi Q^\pi)(s,a,g)]$$

$$= \mathbb{E}_{p(g)\rho_\pi(s,a|g)}[Q^\pi(s,a,g) - \gamma\mathbb{E}_{p(s'|s,a),\pi(a'|s',g)}Q^\pi(s',a',g)]$$

$$= (1-\gamma)\sum_{t=0}^{\infty}\gamma^t\mathbb{E}_{\substack{p(g)\rho_\pi^t(s|g)\\\pi(a|s,g)}}\Big[Q^\pi(s,a,g) - \gamma\mathbb{E}_{\substack{p(s'|s,a)\\\pi(a'|s',g)}}Q^\pi(s',a',g)\Big]$$

$$= (1-\gamma)\sum_{t}\left\{\gamma^t\mathbb{E}_{\substack{p(g)\\\rho_\pi^t(s|g)\\\pi(a|s,g)}}[Q^\pi(s,a,g)] - \gamma^{t+1}\mathbb{E}_{\substack{p(g)\\\rho_\pi^{t+1}(s|g)\\\pi(a|s,g)}}[Q^\pi(s,a,g)]\right\}$$

$$= (1-\gamma)\mathbb{E}_{p(g),\rho^0(s),\pi(a|s,g)}[Q^\pi(s,a,g)]$$

$$= (1-\gamma)\sum_{t}\left\{\gamma^t\mathbb{E}_{\substack{p(g)\\\rho_\mu^t(s)\\\pi(a|s,g)}}[Q^\pi(s,a,g)] - \gamma^{t+1}\mathbb{E}_{\substack{p(g)\\\rho_\mu^{t+1}(s)\\\pi(a|s,g)}}[Q^\pi(s,a,g)]\right\}$$

$$= (1-\gamma)\sum_{t=0}^{\infty}\gamma^t\mathbb{E}_{\substack{p(g)\rho_\mu^t(s,a)\\\pi(\tilde{a}|s,g)}}[Q^\pi(s,\tilde{a},g) - \gamma\mathbb{E}_{\substack{p(s'|s,a)\\\pi(a'|s',g)}}Q^\pi(s',a',g)]$$

$$= \mathbb{E}_{p(g)\rho_\mu(s,a)\pi(\tilde{a}|s,g)}[Q^\pi(s,\tilde{a},g) - \gamma\mathbb{E}_{p(s'|s,a),\pi(a'|s',g)}Q^\pi(s',a',g)]$$

$$\square$$

**Lemma B.2** (Gradient of the noise-contrastive term in energy-based goal-reaching). *Given the following definition for the logit of a NCE-like binary classifier, with* $\rho^+(\overline{g}) = \rho^+(g)$*:*

$$\Delta_\theta(s,a,g,k) = Q_\theta(s,a,g) - \log\mathbb{E}_{\rho^+(\overline{g})}[\exp Q_\theta(s,a,\overline{g})] - \log k \qquad (33)$$

*The gradient of the negative NCE term in the density ratio estimation approaches zero as* $k \to \infty$*:*

$$\frac{d}{d\theta}\mathbb{E}_{\rho_\mu(s,a)\rho^+(g)}\Big[k \cdot \log\Big(1 - \sigma(\Delta_\theta(s,a,g,k))\Big)\Big] \xrightarrow{k\to\infty} 0$$

*Proof of Lemma B.2.* $\sigma$ is the sigmoid function, and $Z_\theta(s,a) = \mathbb{E}_{\rho^+(\overline{g})}[\exp Q_\theta(s,a,\overline{g})]$:

$$1 - \sigma(\Delta_\theta(s,a,g,k)) = \frac{1}{1 + \exp\Delta_\theta(s,a,g,k)} = \frac{1}{1 + \exp(Q_\theta(s,a,g) - \log Z_\theta(s,a))/k} \quad (34)$$

Plugging in the above into the loss and taking the gradient:

$$\frac{d}{d\theta}\mathbb{E}_{\substack{\rho_\mu(s,a)\\\rho^+(g)}}\Big[-k \cdot \log\Big(\frac{\exp Q_\theta(s,a,g)}{k \cdot Z_\theta(s,a)} + 1\Big)\Big] \qquad (35)$$

$$= \frac{d}{d\theta}\mathbb{E}_{\substack{\rho_\mu(s,a)\\\rho^+(g)}}\Big[-k \cdot \log\Big(\frac{1}{k}\exp\Big(Q_\theta(s,a,g) - \log Z_\theta(s,a)\Big) + 1\Big)\Big] \qquad (36)$$

$$= \mathbb{E}_{\substack{\rho_\mu(s,a)\\\rho^+(g)}}\Big[-\frac{\exp Q_\theta(s,a,g)/Z_\theta(s,a)}{1/k \cdot \exp Q_\theta(s,a,g)/Z_\theta(s,a) + 1}\frac{d}{d\theta}\Big(Q_\theta(s,a,g) - \log Z_\theta(s,a)\Big)\Big] \qquad (37)$$

$$\xrightarrow{k\to\infty} \mathbb{E}_{\substack{\rho_\mu(s,a)\\\rho^+(g)}}\Big[-\frac{\exp Q_\theta(s,a,g)}{\mathbb{E}_{\rho^+(\overline{g})}[\exp Q_\theta(s,a,\overline{g})]}\frac{d}{d\theta}\Big(Q_\theta(s,a,g) - \log\mathbb{E}_{\rho^+(\overline{g})}[\exp Q_\theta(s,a,\overline{g})]\Big)\Big] \quad (38)$$

The first term inside the expectation of $\rho_\mu(s, a)$:

$$\frac{-1}{\mathbb{E}_{\rho^+(\overline{g})}[\exp Q_\theta(s, a, \overline{g})]} \mathbb{E}_{\rho^+(g)}[\exp Q_\theta(s, a, g) \frac{d}{d\theta} Q_\theta(s, a, g)] \tag{39}$$

The second term:

$$\frac{\mathbb{E}_{\rho^+(g)}[\exp Q_\theta(s, a, g)]}{\mathbb{E}_{\rho^+(\overline{g})}[\exp Q_\theta(s, a, \overline{g})]} \frac{d}{d\theta} \log \mathbb{E}_{\rho^+(\overline{g})}[\exp Q_\theta(s, a, \overline{g})] \tag{40}$$

$$= \frac{1}{\mathbb{E}_{\rho^+(\overline{g})}[\exp Q_\theta(s, a, \overline{g})]} \frac{d}{d\theta} \mathbb{E}_{\rho^+(\overline{g})}[\exp Q_\theta(s, a, \overline{g})] \tag{41}$$

$$= \frac{1}{\mathbb{E}_{\rho^+(\overline{g})}[\exp Q_\theta(s, a, \overline{g})]} \mathbb{E}_{\rho^+(\overline{g})}[\exp Q_\theta(s, a, \overline{g}) \frac{d}{d\theta} Q_\theta(s, a, \overline{g})] \tag{42}$$

The two terms cancel out and yield a gradient of $0$.

$\square$

**Lemma B.3** (Goal-conditioned Q-functions estimate PMI on given trajectories). *Given that we set apriori the behavioral goal distribution to be $\rho^+(g) = \int_{\mathcal{S} \times \mathcal{A}} p_\mu(s, a, s^+) \mathrm{d}s \mathrm{d}a$. And assuming that the state-action distribution of $\pi$ marginalized over behavioral goals $\mathbb{E}_{\rho^+(g)}[\rho_\pi(s, a \mid g)]$ is the same as $\rho_\mu(s, a)$, which means $\pi$ stays in the same state-action visitation distribution as $\mu$. Then: on trajectories generated by $\mu$, the point-wise mutual information (PMI) between a state-action pair $(s, a)$ and a future state $g$ is given by the Q-function at convergence $Q^\pi$:*

*$PMI((s, a), g) = Q^\pi(s, a, g) - \log \mathbb{E}_{\rho^+(g)}[\exp Q^\pi(s, a, g)] - \log(\overline{r} + (\gamma \mathcal{P}^\pi Q^\pi - Q^\pi)(s, a, g))$*

*Proof of Lemma B.3.* We first start with the optimal $T^*$ in the $f$-divergence bound equation 5, applied to the special case of the function $f$ being a quadratic equation 19:

$$T^*(x) = f'(p(x)/q(x)) \tag{43}$$

$$f'(t) = t - \overline{r} \tag{44}$$

Combining the two, we get:

$$p(x)/q(x) - \overline{r} = T^*(x) \tag{45}$$

Now applying this identity to the $f$-divergence minimization problem in equation 15 (note that we have set $r(s, a, g) = -T(s, a, g)$ in our derivation):

$$\frac{p_\pi(s, a, g)}{p_\mu(s, a, s^+)} = \frac{\rho^+(g)\rho_\pi(s, a \mid g)}{\rho_\mu(s, a)p_\mu^+(g \mid s, a)} = \overline{r} - (Q^\pi - \gamma \mathcal{P}^\pi Q^\pi)(s, a, g) \tag{46}$$

$$= \overline{r} + (\gamma \mathcal{P}^\pi Q^\pi - Q^\pi)(s, a, g) \tag{47}$$

We now take the relationship in equation 21:

$$\frac{\rho_\pi(s, a \mid g)}{\rho_\pi(s, a)} = \frac{\exp Q^\pi(s, a, s^+)}{\mathbb{E}_{\rho^+(g)}[\exp Q^\pi(s, a, g)]} \tag{48}$$

Using this substitution, we arrive at:

$$\frac{\rho^+(g)\rho_\pi(s, a)}{\rho_\mu(s, a)p_\mu^+(g \mid s, a)} \frac{\exp Q^\pi(s, a, s^+)}{\mathbb{E}_{\rho^+(g)}[\exp Q^\pi(s, a, g)]} = \overline{r} + (\gamma \mathcal{P}^\pi Q^\pi - Q^\pi)(s, a, g) \tag{49}$$

Swapping the nominator and denominator and applying the assumption that $\rho_\pi(s, a) = \rho_\mu(s, a)$:

$$\frac{\rho_\mu(s, a)p_\mu^+(g \mid s, a)}{\rho_\mu(s, a)\rho^+(g)} \frac{\mathbb{E}_{\rho^+(g)}[\exp Q^\pi(s, a, g)]}{\exp Q^\pi(s, a, g)} = \frac{1}{\overline{r} + (\gamma \mathcal{P}^\pi Q^\pi - Q^\pi)(s, a, g)} \tag{50}$$

Taking the $\log$ on both sides, we get the following expression of $PMI((s, a), g)$:

$$\log \frac{p_\mu^+(g \mid s, a)}{\rho^+(g)} = Q^\pi(s, a, g) - \log \mathbb{E}_{\rho^+(g)}[\exp Q^\pi(s, a, g)] - \log(\overline{r} + (\gamma \mathcal{P}^\pi Q^\pi - Q^\pi)(s, a, g))$$

$\square$

## B.2 NCE Losses

We now complete the derivation of the NCE losses in section 4.2, by illustrating that the positive classification loss of NCE reduces to an InfoNCE (Van den Oord et al., 2018) loss in equation equation 23 under our definition of the logit equation 22. More specifically, given that we already have Lemma B.2, we only need to show:

$$\frac{d}{d\theta} \log(1 + \exp \Delta_\theta(s, a, s^+, k)) \xrightarrow{k \to \infty} 0$$

We simply follow the definition of $\Delta_\theta$ in equation equation 22 and take the gradient of the above:

$$\frac{\exp \Delta_\theta(s, a, s^+, k)}{1 + \exp \Delta_\theta(s, a, s^+, k)} \frac{d}{d\theta} \Delta_\theta(s, a, s^+, k)$$

$$= \frac{1}{1 + \exp(-\Delta_\theta(s, a, s^+, k))} \left( \nabla_\theta Q_\theta(s, a, s^+) - \frac{\mathbb{E}_{\rho^+(g)}[\exp Q_\theta(s, a, g) \nabla_\theta Q_\theta(s, a, g)]}{\mathbb{E}_{\rho^+(g)}[\exp Q_\theta(s, a, g)]} \right)$$

$$= \frac{1}{1 + k \cdot \frac{\mathbb{E}_{\rho^+(g)}[\exp Q_\theta(s, a, g)]}{\exp Q_\theta(s, a, s^+)}} \left( \nabla_\theta Q_\theta(s, a, s^+) - \frac{\mathbb{E}_{\rho^+(g)}[\exp Q_\theta(s, a, g) \nabla_\theta Q_\theta(s, a, g)]}{\mathbb{E}_{\rho^+(g)}[\exp Q_\theta(s, a, g)]} \right)$$

As $k \to \infty$, we see that the gradient approaches 0 because the scalar on the left approaches 0.

Combining the above result about a part of the positive classification loss in NCE with the Lemma in B.2 which deals with the negative classification loss in NCE, we can arrive at the combined NCE loss at its limit $k \to \infty$, as pointed out in equation 23 of the main text:

$$\arg\max_Q \mathbb{E}_{\rho_\mu(s,a)} \left[ \mathbb{E}_{p_\pi^+(s^+|s,a)}[Q_\theta(s, a, s^+)] - \log \mathbb{E}_{\rho^+(g)}[\exp Q_\theta(s, a, g)] \right]$$

As mentioned, this is roughly equivalent to the InfoNCE loss (Van den Oord et al., 2018), further validating our analysis so far.

## B.3 EBM Losses

To optimize $\mathbb{E}_{\rho_\mu(s,a)}[\mathbb{E}_{p_\pi^+(s^+|s,a)}[Q_\theta(s, a, s^+)] - \log \mathbb{E}_{\rho^+(g)}[\exp Q_\theta(s, a, g)]]$ from an arbitrary dataset of behaviors $\rho_\mu(s, a, s^+)$, we can easily see that the problem lies in accessing $p_\pi^+(s^+ \mid s, a)$: we do not have complete access to the distribution of "positive samples", as sampling directly from $p_\pi^+(s^+ \mid s, a)$ requires on-policy rollouts. But this can be resolved by using importance weights and Equation equation 4 to rewrite $\mathbb{E}_{p_\pi^+(s^+|s,a)}[Q_\theta(s, a, s^+)]$:

$$(1 - \gamma) \cdot \mathbb{E}_{\underset{\rho_\mu(s,a)}{p(s'|s,a)}}[Q_\theta(s, a, s')] + \gamma \cdot \mathbb{E}_{\underset{\rho^+(g)\rho_\mu(s,a)}{p(s'|s,a)\hat{\pi}(a'|s')}} \left[ \frac{p_\pi^+(g \mid s', a')}{\rho^+(g)} Q_\theta(s, a, g) \right] \quad (51)$$

Above, we have introduced a new notation $\hat{\pi}(a \mid s)$ to address the following issue with $p_\pi^+(s^+|s, a)$: while $\pi$ is a goal-conditioned policy, $p_\pi^+(s^+|s, a)$ is not conditioned on a goal *apriori*. A similar problem was encountered (but ignored) in C-Learning (Eysenbach et al., 2020b), which simply assumed that the multi-goal policy was *aposteriori* conditioned on the same future state that got sampled from this policy in the first place (a contradiction). To avoid this problem, we define $\hat{\pi}(a \mid s) = \mathbb{E}_{p^+(g)}[\pi(a \mid s, g)]$ and assume that $p_\pi^+(s^+|s, a)$ is sampled under $\hat{\pi}(a \mid s)$ beyond a single step. We package all the EBM training losses into the following:

$$\mathbb{E}_{\rho_\mu(s,a)} \{ -(1 - \gamma) \cdot \mathbb{E}_{p(s'|s,a)}[Q_\theta(s, a, s')] \} \quad (52)$$

$$\mathbb{E}_{\rho_\mu(s,a)} \left\{ \log \mathbb{E}_{\rho^+(g)}[\exp Q_\theta(s, a, g)] - \gamma \mathbb{E}_{\underset{\rho^+(g)}{\underset{\pi(a'|s',g)}{p(s'|s,a)}}} \left[ \frac{p_\pi^+(g \mid s', a')\hat{\pi}(a' \mid s')}{\rho^+(g)\pi(a' \mid s', g)} Q_\theta(s, a, g) \right] \right\} \quad (53)$$

To decompose the importance weight, note that the policy is trying to maximize the $Q$-values under the entropy constraint in equation 14, resulting in a Boltzmann policy (Haarnoja et al., 2017; Schulman et al., 2017; Haarnoja et al., 2018): $\arg\max_\pi \mathbb{E}_{\pi(a|s,g)}[Q(s, a, g)] - \mathcal{H}(\pi) =$

$\exp Q(s, a, g)/\sum_{\overline{a}} \exp Q(s, \overline{a}, g)$. Minimizing equation 53 is equivalent to minimizing the following loss, with $\perp (\cdot)$ being the stop-gradient sign:

$$\mathbb{E}_{\substack{\rho_\mu(s,a) \\ \rho^+(g) \\ p(s'|s,a)}} \left\{ \perp \left( \frac{\exp Q(s, a, g)}{\mathbb{E}_{\rho^+(\overline{g})}[\exp Q(s, a, \overline{g})]} - \gamma \cdot \frac{\sum_{\overline{a}} \exp Q(s', \overline{a}, g)/|\mathcal{A}|}{\mathbb{E}_{\rho^+(\overline{g})}[\sum_{\overline{a}} \exp Q(s', \overline{a}, \overline{g})/|\mathcal{A}|]} \right) Q_\theta(s, a, g) \right\} \quad (54)$$

*Proof of equation 54.* Let $w_{ebm}(s', a', g)$ denote the importance weight:

$$w_{ebm}(s', a', g) = \frac{p_\pi^+(g \mid s', a')}{\rho^+(g)} \frac{\hat{\pi}(a' \mid s')}{\pi(a' \mid s', g)} \quad (55)$$

Firstly, because of the entropy constraint in equation 14, we have a Boltzmann policy defined on the $Q$-function: $\arg\max_\pi \mathbb{E}_{\pi(a|s,g)}[Q(s, a, g)] - \mathcal{H}(\pi) = \exp Q(s, a, g)/\sum_{\overline{a}} \exp Q(s, \overline{a}, g)$:

$$\pi(a \mid s, g) \propto \exp Q(s, a, g) \quad (56)$$

Secondly, we note that the first term in importance ratio can be estimated from equation 21. Thus we have:

$$w_{ebm}(s', a', g) = \frac{\exp Q(s', a', g)}{\mathbb{E}_{\rho^+(\overline{g})}[\exp Q(s', a', \overline{g})]} \cdot \frac{\mathbb{E}_{\rho^+(\overline{g})}[\exp Q(s', a', \overline{g})]}{\sum_{\overline{a}} \mathbb{E}_{\rho^+(\overline{g})}[\exp Q(s', \overline{a}, \overline{g})]} \cdot \frac{\sum_{\overline{a}} \exp Q(s', \overline{a}, g)}{\exp Q(s', a', g)} \quad (57)$$

$$= \frac{\sum_{\overline{a}} \exp Q(s', \overline{a}, g)}{\sum_{\overline{a}} \mathbb{E}_{\rho^+(\overline{g})}[\exp Q(s', \overline{a}, \overline{g})]} \quad (58)$$

Taking the gradient of equation equation 53 inside the expectation of $\mathbb{E}_{\rho_\mu(s,a)p(s'|s,a)}[\cdot]$:

$$\frac{\mathbb{E}_{\rho^+(g)}[\exp Q(s, a, g) \nabla_\theta Q_\theta(s, a, g)]}{\mathbb{E}_{\rho^+(\overline{g})}[\exp Q(s, a, \overline{g})]} - \frac{\gamma \mathbb{E}_{\rho^+(g)}[\sum_{\overline{a}} \exp Q(s', \overline{a}, g) \nabla_\theta Q_\theta(s, a, g)]}{\sum_{\overline{a}} \mathbb{E}_{\rho^+(\overline{g})}[\exp Q(s', \overline{a}, \overline{g})]} \quad (59)$$

Dividing both the nominator and denominator of the right-hand side by $1/|\mathcal{A}|$, putting the expectation $\mathbb{E}_{\rho_\mu(s,a)p(s'|s,a)}[\cdot]$ back in, and utilizing the stop-gradient sign $\perp (\cdot)$, we arrive at the loss:

$$\mathbb{E}_{\substack{\rho_\mu(s,a) \\ \rho^+(g) \\ p(s'|s,a)}} \left\{ \perp \left( \frac{\exp Q(s, a, g)}{\mathbb{E}_{\rho^+(\overline{g})}[\exp Q(s, a, \overline{g})]} - \gamma \cdot \frac{\sum_{\overline{a}} \exp Q(s', \overline{a}, g)/|\mathcal{A}|}{\mathbb{E}_{\rho^+(\overline{g})}[\sum_{\overline{a}} \exp Q(s', \overline{a}, \overline{g})/|\mathcal{A}|]} \right) Q_\theta(s, a, g) \right\}$$

## B.4 DERIVING HER REWARDS

The purpose of this section is to derive the reward function used in HER from the equation 26:

$$\arg\min_Q \mathbb{E}_{\rho_\mu(s,a)p(s'|s,a)p_\mu^+(s^+|s,a)} \left[ f^*(-(Q_\theta - \gamma \mathcal{P}^\pi Q)(s, a, s^+)) - \beta \cdot (1 - \gamma) Q_\theta(s, a, s') \right]$$

Recall that we have defined $p_\mu^+(s^+ \mid s, a)$ in equation 4:

$$p_\mu^+(s^+ \mid s, a) = (1 - \gamma) p(s^+ \mid s, a) + \gamma \int_{S \times A} p(s' \mid s, a) \mu(a' \mid s') p_\mu^+(s^+ \mid s', a') \mathrm{d}s' \mathrm{d}a'$$

And that we have defined a quadratic form of $f^*$ in equation 19 (with $\overline{r}$ and $\overline{c}$ being constants):

$$f^*(x) = (x + \overline{r})^2/2 + \overline{c}$$

Using the dynamics to expand the expectation and applying the choice of $f^*$ being a quadratic, the loss becomes:

$$\arg\min_Q \mathbb{E}_{\rho_\mu(s,a)p(s'|s,a)}(1 - \gamma) \cdot \left[ \frac{1}{2}\left(\overline{r} + (\gamma \mathcal{P}^\pi Q - Q_\theta)(s, a, s')\right)^2 - \beta \cdot Q_\theta(s, a, s') \right]$$

$$+ \mathbb{E}_{\rho_\mu(s,a)p(s'|s,a)\mu(a'|s')p_\mu^+(s^+|s',a')} \left[ \gamma \cdot \frac{1}{2}\left(\overline{r} + (\gamma \mathcal{P}^\pi Q - Q_\theta)(s, a, s^+)\right)^2 \right]$$

*Assuming that there is a stop gradient sign on $\mathcal{P}^\pi Q$ because of the use of a target network*, we can rewrite the above as one single quadratic and see that the gradient of the above loss w.r.t $Q$ is equivalent to the gradient of the following squared Bellman residual:

$$\arg\min_Q \mathbb{E}_{\rho_\mu(s,a)p(s'|s,a)p_\mu^+(s^+|s,a)}\left[\frac{1}{2}\Big(r(s,a,s',s^+) + (\gamma\mathcal{P}^\pi Q - Q_\theta)(s,a,s^+)\Big)^2\right]$$

where the reward function $r(s, a, s', s^+)$ is:

$$\begin{cases} \overline{r} + \beta, & s' = s^+ \\ \overline{r}, & s' \neq s^+ \end{cases}$$

$\square$

## C  ABLATIONS AND HYPER-PARAMETERS

In this section, we include additional ablations and hyper-parameters. We first describe the goal-reaching environments in more details in Figure 9. We use the following thresholds (for Euclidean norms) for determining success: $[0.08, 0.08, 0.05, 0.05, 0.1]$, which are tight thresholds based on our visualizations of the environments. We use the same network architecture, sampling and optimization schedules for all the methods, as described in Table 2. As for $\gamma_{\text{HDM}}$, we set it to be $0.85$ in *Four Rooms* and *Lunar Lander*, $0.5$ in *Sawyer Push* and *Claw Manipulate*, and $0.4$ for *Door Opening*. Ablation on this hyper-parameter can be found in Figure 10. For next state relabeling ratio, we set the default to be $0.2$, increase it to $0.5$ in *Lunar Lander*, and $0.6$ in *Sawyer Push* and *Door Opening*. For the soft-$Q$-learning (Schulman et al., 2017) + HER baseline, we set the temperature parameter to be $0.2$, which we have found to empirically perform well.

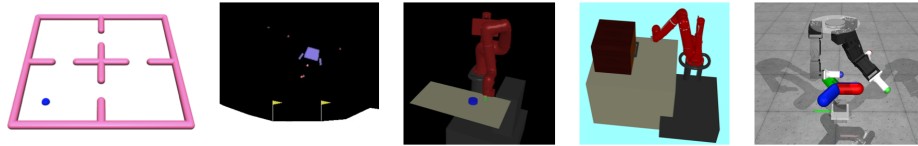

Figure 9: Goal-reaching environments from GCSL (Ghosh et al., 2019) that we consider in this paper: reaching a goal location in *Four Rooms*, landing at a goal location in *Lunar Lander*, pushing a puck to a goal location in *Sawyer Push*, opening the door to a goal angle in *Door Open* (Nair et al., 2018b), turning a valve to a goal orientation in *Claw Manipulate* (Ahn et al., 2020).

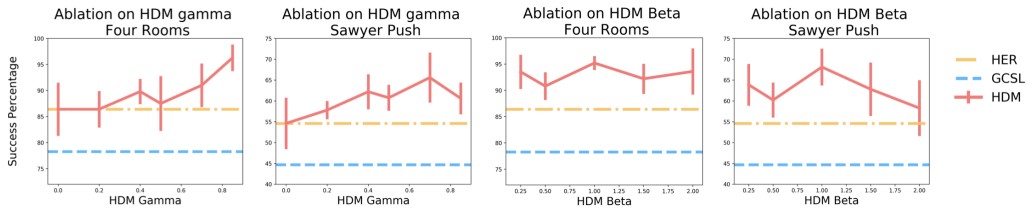

Figure 10: Ablation studies on HDM Gamma and Beta. HDM Gamma refers to $\gamma_{\text{hdm}}$ in equation 31, and HDM Beta refers to the $\beta$ term in equation 14. The orange line and the blue line denote HER and GCSL baseline performance. See Section 5.2 for further discussion of these results.

Table 2: Hyper-parameters

| Parameters | Value |
|---|---|
| Optimizer | Adam (Kingma & Ba, 2014) |
| Number of hidden layers (all networks) | 2 |
| Number of hidden units per layer | [400, 300] (Fujimoto et al., 2018) |
| Non-linearity | ReLU |
| Polyak for target network | 0.995 |
| Target update interval | 10 |
| Ratio between env vs optimization steps | 1 |
| Initial random trajectories | 200 |
| Hindsight relabelling ratio | 0.85 |
| Update every # of steps in environment | 50 |
| Next state relabelling ratio | 0.2 |
| Learning rate | 5.e-4 |
| Batch size | 256 |

