# OpenReview forum: "Understanding Hindsight Goal Relabeling Requires Rethinking Divergence Minimization"
_ICLR.cc/2023/Conference — Submitted to ICLR 2023_

### Official Review · Reviewer_CKyn · 2022-10-21

**Confidence:** 4
**Correctness:** 2
**Technical Novelty And Significance:** 2
**Empirical Novelty And Significance:** 3
**Recommendation:** 3

**Clarity, Quality, Novelty And Reproducibility:**

**Clarity** -- I have a number of concerns about the clarity of the paper (see above).

**Quality** -- I have a number of concerns about the correctness of the paper (see above).

**Novelty** -- The precise form of the method is novel, to the best of my knowledge. It is very similar to a number of prior methods, and I don't think these similarities are adequately discussed.

**Reproducibility** -- The paper includes code, along with launch scripts to reproduce one of the experiments.

**Strength And Weaknesses:**

Strengths
* The empirical results seem quite strong.

Weaknesses
* I am unsure about the motivation for the paper. The introduction seems to argue that there aren't any good ways of defining the goal-conditioned RL problem without reward engineering, but I'm not sure this is true (see below). The proposed objective is a combination of a "goal likelihood term" (which is the same as or quite similar to uncited prior work [5, 6, 8]) and an f-divergence term (from prior work, and cited). Most of the paper's contributions seem to be about analyzing the f-divergence term, but it is unclear why that term is needed at all: directly maximizing the goal likelihood term would yield the optimal goal reaching policy. Note that prior work has even considered maximizing the combination of the goal likelihood term and the f-divergence term, albeit with a different choice of f-divergence [9].
* I am unsure about some of the claims in the introduction (see below)
* I am unsure about certain parts of the math. It's possible they are correct and I was just confused by the notation (see details below).
* Many parts of the paper were difficult to follow (see below).
* The proposed method seems quite complex.

Detailed comments
* "this connection between imitation and hindsight relabeling is not well understood" -- I'm not sure this is true. There's been a fair bit of analysis of this in prior work (e.g., [1, 2, 3]).
* "reward function in hindsight experience replay" -- Where is this shown? The HER paper uses a sparse reward on human-specified coordinates.
* "without reward engineering" -- All four of these papers *do* require reward engineering, in the form of specifying a distance metric and (for some) specifying coordinates of interest. Consider referring to prior work that actually does learning without reward engineering [5, 6, 7].
* "it is unclear how to write down a well-defined objective for goal-conditioned policies" -- I don't think this is true. See, e.g., [5, 6].
* "known as contrastive divergence" -- I would recommend *not* referring to this as contrastive divergence, which typically refers to a specific approximate algorithm for fitting EBMs (instead of generating true samples from the model, it approximates the samples using a single gradient step).
* "NCE gives EBM a more tractable way to maximize data likelihood" -- I didn't understand this. Would it be possible to elaborate?
* Eq 10, Eq 11 -- I found these equations and the surrounding discussion confusing. Are these distributions supposed to be the same?
* Eq 12 -- There seems to be a "slight-of-hand" here, where the future state is reinterpreted as the commanded goal. I'd recommend clarifying this by explicitly writing down the distributions for the relevant random variables ($g$, $s^+$).
* Eq. 15 -- I'd recommend making sure the arguments to the f-divergence have the same arguments; currently, one has $g$ and one has $s^+$.
* Eq 15 -- How does this relate to Eq 12? It seems like the direction of the KL has been silently reversed
* Eq 15 -- Is this f-divergence being optimized? If so, w.r.t. what components?
* Eq 16 -- How is $\rho_\pi(s, a, \mid g)$ defined? Is this equal to $\rho_{\pi(a \mid s)}(s, a \mid s^+=g)$?
* Lemma 4.1 -- What reward function is used for defining $Q^\pi(s, a, g)$?
* "now define a Q-function as ... separate from the one defined in Eq. 18" -- I found this confusing. I'd recommend using the standard convention of defining Q-function as expected discounted returns, and using a different variable for this.
* Lemma 4.2 -- I found this hard to parse. I'd recommend adding scaffolding: what are the main aims of the result, what will this result be used to show?
* Eq. 25: How is this similar to and different from [6]?
* Eq 25 -- Can this be re-expressed without the Q-functions, directly in terms of the environment dynamics?
* Eq. 28 -- I'm not sure $s' = s^+$ is well defined for continuous settings.
* "EBM training equation 53" -- Broken reference?
* Fig 8 -- I was unable to read the text in this figure.
* Table 1 -- How are the numbers for GCSL computed? [10] reports much higher numbers for a GCSL-style method on the door task (This paper reports GCSL = 19% and the proposed method gets 87%, but [10] reports that a GCSL-style method gets 95.8%).
* code, scripts folder -- Why are different random seeds used for different environments.
* logger.py, L480 -- I'd recommend not including expletives in open source code.

[1] https://proceedings.neurips.cc/paper/2019/file/3891b14b5d8cce2fdd8dcdb4ded28f6d-Paper.pdf

[2] http://proceedings.mlr.press/v130/tang21b/tang21b.pdf

[3] https://proceedings.neurips.cc/paper/2020/file/a97da629b098b75c294dffdc3e463904-Paper.pdf

[4] https://arxiv.org/pdf/1912.06088.pdf

[5] https://arxiv.org/pdf/2101.07123.pdf

[6] https://arxiv.org/pdf/2011.08909.pdf

[7] https://arxiv.org/pdf/1905.07866.pdf

[8] https://arxiv.org/abs/2104.10190

[9] https://arxiv.org/pdf/2206.07568.pdf

[10] https://arxiv.org/pdf/2112.10751.pdf

**Summary Of The Paper:**

This paper proposes a new method for goal-conditioned RL that combines three terms: (1) goal-conditioned behavioral cloning, (2) maximizing the likelihood of the goal, and (3) an exploration term. The paper proposes to use a different f-divergence for optimizing the first term, which ends up being done using a DICE-like TD update. Empirically, the proposed method outperforms baselines on a goal-reaching benchmark from prior work.

**Summary Of The Review:**

My large number of concerns about the paper compel me to vote to reject it. I do think there might be some really interesting ideas in the TD-like method for minimizing the f-divergence. For example, might that allow a GCSL/HBC-style method to perform off-policy learning?

---

> ### Author Response · Authors · 2022-11-15
> **Author Response (Part I)**
>
> We thank the Reviewer for the feedback. Unfortunately, it seems there were several fundamental misunderstandings of our work. We are not entirely sure why this happened, but hopefully we can clarify things with our rebuttal.
>
> To begin with, the provided Summary of The Paper is incorrect. Notably, it deviates not only from the paper, but from the summaries of all three other reviewers as well. The Reviewer states that the paper proposes “a new method” for goal-reaching that combines three terms, and its novelty lies in “using a different f-divergence for optimizing the first term”. However, this analysis completely misses the main point of the paper.  **The main objective we proposed was not a “new method”, but rather a unified framework for existing methods.** Moreover, we did not use a different f-divergence at all; what we showed was that existing methods are optimizing a specific f-divergence, and thus a special case of our proposed framework. The other three reviewers did seem to understand this point. And all feedback we have thus far received regarding this paper suggests this point is clear. Given how clear this was made in the abstract and introduction, we struggle to think of how the clarity might be improved further here.
>
> Much of the Reviewer’s subsequent confusion appears to stem from the fundamental misinterpretation above. For instance, the Reviewer stated that “most of the paper's contributions seem to be about analyzing the f-divergence term, but it is unclear why that term is needed at all”. This is perplexing. **The goal of analyzing the f-divergence term was to provide a first-principled derivation of the reward function in Hindsight Experience Replay (HER).** We have repeatedly stated this goal and the significance of this result throughout the paper, but this part of the contribution was completely ignored and declared to be unnecessary by the Reviewer. Why was this ignored? You can not throw out a paper’s main contribution in such a deliberate manner.
>
> Many aspects of the given Review are very puzzling:
>  - The Reviewer commented that “the proposed method seems quite complex.” The proposed method is a 5-line algorithm in the Algorithm box. The theoretical unification of many existing algorithms is indeed complex, but our final method itself is very simple. The provided code also shows how this method can be implemented with minimal difficulty over previous algorithms in this space.
>  - The paper has already included a comprehensive table for notations in the Appendix on page 14. Most of the Reviewer’s questions about notations can be directly found in the notation table, and we have made an effort to make the notations in the main paper self-contained as well.
>  - The Reviewer consistently ignores the papers that we have already cited, and then claims that we didn’t cite them. **We have already cited [3, 4, 6, 7].** Regarding other papers that we are asked to cite, the Reviewer claims that [5] also defined a “goal-likelihood term”, but we cannot find any term remotely similar in its 100-page document. [8] is not a model-free method because it requires fitting a dynamics model and then using its log-likelihood as the per-step reward. [1] is only shown to work on Grid-World and Fetch-Reach. [2] is the same as HBC, and does not rigorously define what inferring “optimality” means for the goal-conditioned RL. [10] studies offline RL. We are happy to provide a citation for [9], but note that it is a concurrent work in NeurIPS and a special case of C-learning (which we have already cited).
>  - Many parts of the Review are self-contradictory. For instance, the Reviewer claims that HER does reward-engineering but C-Learning does not do reward-engineering, but those two algorithms are both goal-reaching algorithms and study the exact same setting. The reviewer said that Eq 28 (the HER reward) is “not well defined for continuous settings”, while also stating that “it is unclear why [f-divergence] term is needed at all”, not realizing that it is exactly the f-divergence term that makes this reward well-defined for continuous settings.
>  - The purpose of the provided script folder in the code is to illustrate how to run the code with a single seed, rather than how to launch a batch of experiments to the training cluster with a list of different seeds to get the variance of success rates. **We have reported the variance of the results under five seeds, both for the baselines and for our method. In contrast, the results that the Reviewer linked to in [10] did not report any variance in their results.** This is particularly remarkable considering that GCSL-like methods suffer from high variance, as we have shown in our experiments.

---

> > ### Author Response · Authors · 2022-11-15
> > **Author Response (Part II)**
> >
> > Continuing our previous response:
> >  - The Reviewer raised an issue with a unit test function in our logger.py. We simply built on top of OpenAI Baselines and Stable Baselines (the two most popular RL codebases today), and **the logger.py is the same as the one in those two open-sourced codebases.** If the Reviewer insists, we would encourage the Reviewer to raise the concern about this particular unit test on the Github repositories of those two codebases.
> >
> > Given that the Review seemed to have completely misunderstood the core contribution of our work, and that the Rating of 1 is in clear disagreement with all other Reviews, we respectfully disagree with the assessment.

---

> > ### Comment · Reviewer_CKyn · 2022-11-15
> > **Reviewer response**
> >
> > Dear authors,
> >
> > Thank you for the detailed response, and for all of the clarifications!
> >
> > I agree that more of the paper discusses the theoretical unification, rather than the actually proposed method. I opted to focus my review on the proposed method and the experiments not because I missed that part of the paper, but because I think the experimental results are more significant (and because I dispute many of the claims in the theoretical section). I wrote the review to answer the question: ``what changes to the paper would make me inclined to accept and advocate for the paper?''
> >
> > Broadly, my concerns about the relationship to prior work were not that prior work had proposed exactly the same method, but that it had already developed the key elements (but, perhaps, with a different underlying RL algorithms). Clarifying a few of the comments above:
> >
> > > the proposed method seems quite complex
> >
> > I was thinking about Eq. 14 here. My understanding is that there has been prior work that studies (a), work that studies (b), and work that studies (c). So, while adding all the objectives together does mean that they are unified, it also means that the resulting method must be at least as complex as any of the constituent components.
> >
> > > The paper has already included a comprehensive table for notations in the Appendix on page 14
> >
> > I read the main paper before reading the appendix. So, when reading the main paper, I was confused about notation that was only later defined in the Appendix.
> >
> > > We have already cited [3, 4, 6, 7].
> >
> > I felt like the discussion of these papers in the main text weren't accurate, which is why I brought them up in the review.
> >
> > > [5] also defined a “goal-likelihood term”, but we cannot find any term remotely similar in its 100-page document.
> >
> > See, e.g., Eq. 7. The matrix $M$ encodes the probability of reaching states at some point in the future.

---

### Official Review · Reviewer_4ceU · 2022-10-23

**Confidence:** 2
**Correctness:** 4
**Technical Novelty And Significance:** 3
**Empirical Novelty And Significance:** 3
**Recommendation:** 6

**Clarity, Quality, Novelty And Reproducibility:**

I found the work clear in sections, but as listed above, there are some areas in the communication that were not entirely obvious to me. Having said this, although I didn't spend significant time re-deriving the formulae, as far as I can see the equations seem to follow. Regarding originality, I am not an expert on GCRL, but the framework presented seems novel enough to merit interest. For reproducibility, pseudo-code and hyperparameters are given, and code is provided.

**Strength And Weaknesses:**

Strengths:
* The unifying framework appears useful, and I found the unification of HER under a goal-conditioned RL framework interesting, and appears to provide insight into how these methods may work on a more theoretical level.
* The authors provide empirical evidence that their approach HDM, which balances goal reaching and state-action matching, outperforms prior methods which don't explicitly manage this effectively.

Weaknesses:
* While the paper is easy to follow in sections, I found the overall narrative a bit unclear. For instance, Section 3 was quite hard to follow, as quantities and PGMs/distributions are introduced with not much intuition, for instance Eqs 10,11 and 12. Explaining the quantities in Eq 10 and 11, and why 12 gives HBC/GCSL would help here. Similarly, when the authors present their approach in Eq 14, the divergence term is flipped, but no explanation as to why this is done appears to be provided. Another case is when deriving the HER objective under their f-divergence formulation; they seem to arbitrarily combine the 1-step part of the EBM formulation with part of the f-divergence term, but don't give explanation why replacing the multi-step part of the EBM objective like this is fine? I also was a bit confused by the inclusion of 4.3 (HER derivation) and 4.4 (BC analysis) in section 4; it might make more sense to separate them (e.g., Section 4 being more focused on just the creation of the optimizable objective, and another section showing how their framework can unify these approaches).
* I wasn't entirely convinced by some of the ablations. For instance, in the appendix the authors ablate values of Beta and Gamma; wouldn't we expect some extreme settings to recover the performance of HBC/GCSL? If not, why?
* Nits:
   * The term Bellman Residual is somewhat confusing, as it is overloaded with Bellman Residual learning [1]; perhaps using Bellman Error would be sufficient here, as I believe the authors explicitly state they put a stop gradient on the target values.
   * How many seeds are used in the experiments?

[1] Residual algorithms: Reinforcement learning with function approximation, Baird, ICML 1995

**Summary Of The Paper:**

In this work, the authors present a unification of goal-conditioned behavior cloning/supervised learning and hindsight relabeling methods (e.g., HER) under the framework of divergence minimization. They show that HER is a special case of a divergence minimization when considered under a goal-conditioned Q-learning approach (when Q is treated as an EBM).

To this end, they propose a novel objective that aims to account for and balance the key elements required in goal-conditioned RL, and show promising results in a suite of standard tasks.

They further provide intuition about how excluding some of these elements (e.g., lack of goal-likelihood in the optimization) results in sub-optimal performance, and use this framework as a means to explain the poorer performance of pre-existing approaches.

**Summary Of The Review:**

Overall, I believe this paper merits an acceptance. I think the work sheds interesting light on GCRL, particularly where approaches like HER fit in, and the empirical results seem to largely back up the intuitions presented. As a result, I believe the paper is of interest to the community, and is novel enough in its analysis to merit acceptance at a major conference.

I have some issues with the presentation and experiments that I have listed in the Weaknesses section, and would be grateful to hear from the authors about addressing these.

---

> ### Author Response · Authors · 2022-11-16
> **Author Response**
>
> Dear Reviewer,
>
> Thank you for your review!
>
> Overall, it seems that the reviewer agrees that the presented framework and methodology are of interest to the community (“the work sheds interesting light on GCRL, particularly where approaches like HER fit in, and the empirical results seem to largely back up the intuitions presented”, “the framework presented seems novel enough to merit interest”), and the main concerns are about certain parts of the expositions regarding PGMs, the order of f-divergence, and the combination of f-divergence minimization and EBM. We will address those questions one by one:
>  - “PGMs/distributions are introduced with not much intuition, for instance Eqs 10, 11 and 12.”: Eq 10 is the hindsight relabeled distribution. It can be thought of as (sub-optimal) expert demonstrations. It combines the given state-action visitation distribution and the future states along the same trajectories, thus providing demonstrations of how to reach those future states as goals. It is the distribution that both Eq 11 and Eq 13 are trying to match / “imitate”. Eq 12, like BC, does not match the states, and only matches state-conditioned action distributions. Not matching the states would lead to a compounding of errors in sequential decision making, as analyzed in DAGGER. Eq 12 is a straightforward application of KL divergence definition, but we will include more details about this in the appendix of the final draft. The Eq 12 objective is basically BC but conditioned on hindsight goals.
>  - “When the authors present their approach in Eq 14, the divergence term is flipped”: we flip the divergence term so that Lemma 4.1 can be applied. After flipping the divergence term, we arrive at Eq 14, where the terms for the “online rollout” component are linear. This is a crucial condition for the online-to-offline transformation (Lemma 4.1) to be applied. Flipping the divergence term is required for the math to work out.
>  - “Why replace the multi-step part of the EBM objective” in HER reward: Section 4.3 shows that, by ignoring the multi-step part of the EBM objective, we can derive the HER reward. Section 4.4 analyzes the ignored multi-step part of the EBM objective and derives an additional loss to “correct” the HER reward. The analysis shows that the HER reward might not be doing the entirely “correct” thing. Nevertheless, in Section 4.4, we discussed why the HER reward alone still worked reasonably well: “by not pushing up the Q-values of the discounted future-state distribution beyond a single step, the HER agent is encouraged to reach a goal sooner rather than later.”
>  - All experiments used five seeds. We have also already provided the algorithm box, the hyper-parameters, and the code.
>  - Whether HBC/GCSL can be recovered from extreme values of Beta and Gamma: HBC only optimizes the first term in Eq 14, and does not do state-matching. For HDM, even if we set Beta to 0, and thus only optimize the first term of Eq 14 in HDM, HDM would still do state matching, so what those two approaches optimize would still be fundamentally different. See the difference between Fig 3 and Fig 4. Additionally, the Q-learning-like learning procedure in Eq 18 based on an L2 loss and a target network tends to behave quite differently from a BC-like maximum likelihood loss on the actions (Eq 12).
>
> We hope that our response above has clarified the reviewer’s remaining concerns about the paper.

---

### Official Review · Reviewer_jwR4 · 2022-10-24

**Confidence:** 2
**Correctness:** 3
**Technical Novelty And Significance:** 3
**Empirical Novelty And Significance:** 3
**Recommendation:** 6

**Clarity, Quality, Novelty And Reproducibility:**

As discussed above, clarity may be among the most crucial problem. If I have a guess, the authors tried to be highly comprehensive and rigorous by referring to the equations and the related literature while providing intuitions simultaneously.
Unfortunately, it was too much for me (Maybe, I am not strong enough in the field, and a more advanced reader may have a different perspective). Anyway, I think that better decoupling the concepts (link to other algorithms, proof, intuitions) would greatly ease the reading.
Another unfortunate consequence is that it also becomes less clear what are the actual contributions and core differences between the methods and the literature. The writing sometimes gives the (misleading) feeling that the authors did a patchwork of many algorithms in their method. For instance, it would be easier to show that other methods may be derivate from their training objectives.
In a few words, I would encourage the author to make the paper more digest by: better disentangling the concepts and moving some parts in the appendix to make the paper less dense.

On a different note, I found the introduction a bit abrupt, and there is a few over-statement from my perspective. Reward is not enough because they are manually created is quite a shortcut. Why is divergence minimization the de facto way to describe imitation?

**Strength And Weaknesses:**

The main strength of this paper is to formalize many algorithms into a single consistent framework and draw links between the methods. It also results in a clear objective and a practical algorithm. Unfortunately, I have difficulty grasping clear intuitions by following mathematical explanations. Too many mathematical references are interleaved in a few steps, and the notation sometimes feels too complex with too many nuances. For instance, Section 3 is a good example; there are many mental jumps to reach the final objective. On a single page, there are 4 figures; the authors refer to 4 different algorithms and introduce 5 equations with hard-to-parse notations $p_\nu$ vs. $\rho_\nu$). I believe the authors wanted to be too comprehensive and highly rigorous, but it sometimes makes things too dense.

The final empirical results are also quite promising. However, I would recommend the authors provide more experimental details to be fully convincing (The appendix is a bit short on the matter.) First, I am not sure whether the number of seeds was provided. Then, training curves would be a nice add-on. Similarly, it is unclear whether the authors reproduced the results from other algorithms or whether they reported the scores (hence having potentially different training hyperparameters). Finally, I would have enjoyed more discussions on the hyperparameters (not only that they outperform classic methods on an extensive range). Idem testing the algorithm with different r and beta, for example, (-1,0) vs. (0, 1). What are the extreme case, how should they be set, etc.? I understand that the paper's primary focus is not on the empirical sections, but the paper could be greatly strengthened. The comparisons between the different baselines could be made more comprehensive. A simple table to highlight core elements (reward, clipping, etc.) would be helpful

Then, I have a few other remarks:
 - In the algorithm, one step-q-learning seems like a significant constraint (l1), especially with a long horizon. Why not have a lambda/n-step return
- How n-step return would behave with $\gamma_{hdm}$
- Would it make sense to use DQfD[1] instead of pure BC?
- There is no Bellman residual error here, it is only a TD error

[1] Hester, Todd, et al. "Deep q-learning from demonstrations." Proceedings of the AAAI Conference on Artificial Intelligence. Vol. 32. No. 1. 2018.

**Summary Of The Paper:**

This paper casts the problem of hindsight goal relabeling into a divergence minimization problem.
The authors propose a simple loss form to optimize and provide an algorithm to efficiently mix behavior cloning and RL in the goal relabelling procedure.

To do so, the authors first summarize the many mathematical tools they used in the paper (Rl, Divergence Minimization, EBM). They en combine those concepts to finally explore how to maximize this objective.  Their derivations also allow the author to see when BC may help improve the policy (as some trajectories may be sub-optimal).

Finally, the authors perform a short empirical study over classic goal relabelling tasks and empirically show that their approach consistently outperforms other goal relabelling methods.

**Summary Of The Review:**

I think this paper can be an exciting piece of work that completes the goal of relabelling the framework.
However, the lack of clarity and some issues in the experimental results prevent me from being confident in my assessment. There are still many points that may remain obscure to me. Besides, the cluttered notation was really painful and prevented me from digging much into the mathematical details. Therefore, I am leaning to a borderline accept with little confidence

---

> ### Author Response · Authors · 2022-11-16
> **Author Response**
>
> Thank you for your review! We are glad that the reviewer recognized the technical contributions of the paper, and agreed that the contributions could be of interest to the community (“The main strength of this paper is to formalize many algorithms into a single consistent framework and draw links between the methods. It also results in a clear objective and a practical algorithm”; “this paper can be an exciting piece of work that completes the goal relabelling framework.”) The two main concerns seem to be: 1. Writing being too dense; 2. More experimental details and more experiments.
>
> Firstly, regarding the “writing being too dense” issue, the reviewer suggested that “better decoupling the concepts (link to other algorithms, proof, intuitions) would greatly ease the reading”. We apologize for the writing of the paper being condensed due to the inherent page limit of a conference paper. Given that all the proofs and most of the details are already in the Appendix, what additional sections would the reviewer recommend moving to the Appendix? In our view, further shrinking and removing existing sections in the main paper might cut off the logical flow of the paper. We have already made an effort in the paper to decouple and clarify things: for instance, Section 3 only discusses *what* objectives to optimize, and Section 4 only discusses *how* to optimize the proposed objective. Different sub-sections in Section 4 deal with different terms in the objective. At the end of Section 4, we have a 5-line Algorithm box to remind the readers what the method does. **We are open to concrete suggestions on how to make the writing less dense, but we strongly feel that the difficulty stems almost entirely from the scope of the paper** - bridging together notations and ideas from three separate areas of learning to derive a divergence minimization framework for HER required a large amount of machinery.
>
> Secondly, regarding the experiments: we are happy to provide more experimental details (note that we have also provided the code). All experiments used five seeds, and we presented the results in a table format because it more clearly shows the difference in performances. We reproduced the results from other algorithms under reported hyper-parameters. We have already shown results for HER with both (-1, 0) and (0, 1) rewards, and we have remarked in the experimental section that “those two types of rewards lead to different learning dynamics because the Q-function is initialized to output values around 0.” It seems that HER with (-1, 0) reward works slightly better, so we have set that to be the default in our Algorithm box. We have provided the ablation studies on the only two additional hyper-parameters that we introduce. The reviewer also mentioned methods like multi-step Q-learning loss and DQfD and asked if those methods could be used. Besides the fact that those methods have their own complications (for instance, multi-step Q-learning often requires importance-weight correction), introducing more moving parts to the method would further complicate the question of what helped the performance and what did not, and those experiments are out of the scope of what we study in this paper. The purpose of our experimental section is to further validate our proposed framework, not necessarily achieve SOTA results. We believe our experiments have sufficiently demonstrated the usefulness of our proposed framework and method.
>
> The reviewer remarked that “the writing sometimes gives the (misleading) feeling that the authors did a patchwork of many algorithms in their method”. Indeed, the proposed approach is very simple, does not introduce any additional moving parts, and only requires minimal changes on top of existing algorithms. Although those factors might give the impression that the proposed method did a “patchwork”, we actually think that the simplicity is a strength, rather than a weakness, of our method. **Our mathematical derivation might be complex, but our proposed algorithmic change is pretty straightforward.**
>
> We hope that our response above has clarified the reviewer’s remaining concerns about the paper.

---

### Official Review · Reviewer_aKS5 · 2022-10-25

**Confidence:** 4
**Correctness:** 4
**Technical Novelty And Significance:** 3
**Empirical Novelty And Significance:** 2
**Recommendation:** 5

**Clarity, Quality, Novelty And Reproducibility:**

* Clarity and Quality: Fair. As pointed out above, the mathiness retracts from ease of following the exposition, but the notation and math seems clear enough.
* Novelty: Weak. It is unclear how this approach compares and contrasts with prior work. There is no dedicated related work comparison, and significant recent work has not been addressed.
* Reproducibility: Good: Setup and hyper-parameters have been provided.

**Strength And Weaknesses:**

### Strengths
* This paper conducts a study of why hindsight replay helps when learning in goal-conditioned RL from the perspective of divergence minimization.
* This study leads the paper to propose an adaptive algorithm that can take advantage of high quality demonstrations.
* The perspective of goal-conditioned RL as a form of divergence minimization could be interesting.

### Questions and Weaknesses:
* The paper does not have a consolidated related works section. Are there other papers that aim to address goal-conditioned RL via a divergence minimization perspective? For example, it seems that the paper should compare to [1,2] in their exposition as well as their experiments.
* While the rigor in section 4 is to be appreciated, it takes away from ease of understanding and clarity.
* Section 5.3 was unclear on what _achieved-goal_ means. If that section essentially points out that GCSL works better in situations where the trajectories in the dataset get to the goal efficiently, then it is unclear why this is an important insight. BC with expert trajectories is more likely to work better. A better comparison would have been to also plot HER + HBC to show that these sorts of trajectories do not hard HER learning as much.
* How do the results in Table 1 differ from the results in Figure 7?

### References:
[1] Ma, Y.J., Yan, J., Jayaraman, D. and Bastani, O., 2022. How Far I'll Go: Offline Goal-Conditioned Reinforcement Learning via $ f $-Advantage Regression. arXiv preprint arXiv:2206.03023.

[2] Durugkar, I., Tec, M., Niekum, S. and Stone, P., 2021. Adversarial intrinsic motivation for reinforcement learning. Advances in Neural Information Processing Systems, 34, pp.8622-8636.

**Summary Of The Paper:**

This paper analyzes hindsight experience replay (HER) through divergence minimization with energy based models. With this analysis, it presents an approach that combines HER with behavioral cloning (BC) by only imitating actions in the dataset if they move the agent a certain amount towards the goal.

The paper concludes by comparing this adaptive HER + BC technique (termed hindsight divergence minimization, HDM) to HER, behavioral cloning and a more straightforward implementation of HER +BC.
The paper concludes with a hypothesis that BC-based techniques work better if the dataset used for learning has a higher proportion of achieved goals in its trajectories, and validates this hypothesis via experiments.

**Summary Of The Review:**

The paper offers an interesting perspective on HER, and presents an approach to adaptively improve performance by incorporating BC into the objective if it is helping the agent learn. However, there might be related work that is not addressed in the paper, and might perhaps need to be compared to.

---

> ### Author Response · Authors · 2022-11-15
> **Author Response**
>
> Thank you for your review! Overall, the review seems to agree that this work makes a solid technical contribution and could be of interest to the community (“The perspective of goal-conditioned RL as a form of divergence minimization could be interesting”; “The paper offers an interesting perspective on HER, and presents an approach to adaptively improve performance by incorporating BC into the objective if it is helping the agent learn.”) The two main concerns seem to be that the paper does not have a separate related works section, and that the paper did not compare to the two baselines that the Reviewer proposed.
>
> Regarding related works, **our paper has already extensively cited a wide range of papers across the fields of goal-conditioned RL, inverse RL, and energy-based models. Our references already have 4 pages.** We thank the reviewer for pointing out two references that we missed, and we will cite the two additional papers that the Reviewer referred to. The paper has merged the related works section into the Background section (Section 2) and the graphical model comparison section (Section 3). We presented the related works in this way to better illustrate the nuances between different methods.
>
> Regarding the two baselines that the reviewer asked us to compare to, **we will cite those two papers in the final draft**, but we believe that comparing with those two additional baselines will confound rather than strengthen the goal of the paper.
>  - The “How Far I'll Go” paper [1] proposes **method GoFAR that “does not require any hindsight relabeling”. In contrast, the purpose of our paper is to study why hindsight relabeling works in the first place,** and all our baselines use various forms of hindsight relabeling to show the readers which hindsight-based learning method works best.
>  - The “Adversarial intrinsic motivation for reinforcement learning” paper [2] studies intrinsic motivation and exploration strategies. **Exploration is orthogonal to what we study in this paper. Indeed, we can apply additional exploration strategies to not just our method but also all our baselines.** Moreover, this “adversarial intrinsic motivation” method requires training a GAIL-like discriminator as the reward. It introduces an additional moving part, and adversarial training tends to be unstable. Our method and all our baselines do not require adversarial training or an additional learned reward module.
>
> Regarding other questions from the reviewer: “achieved goal” means the future states within the same trajectory (which are the goals used for hindsight relabeling); results in Figure 7 are the same as the results in Table 1 but with the performance normalized over GCSL to highlight the percentage of performance gain. We agree with the reviewer that “BC with expert trajectories is more likely to work better”, but in self-supervised goal reaching, most trajectories are sub-optimal, and vanilla BC-like methods struggle to deal with this kind of sub-optimality.
>
> Overall, it seems that you were quite worried about the lack of discussion on related work making it difficult to judge the paper’s novelty. Could you please let us know if the above comments are sufficient to alleviate those concerns? If not, can you please make further suggestions on what comparisons you would like to see and what work you would like discussed?

---

> ### Author Response · Authors · 2022-11-28
> **Looking Forward to Reviewer Response**
>
> Dear Reviewer aKS5,
>
> This is a gentle reminder that we have posted an author response regarding your concerns about Related Works. We believe that we have extensively cited related works, and have made sufficient experimental comparisons to support the main goal of the paper (which is to better understand a family of existing methods). We will also cite the two additional papers, GoFAR [1] and AIM [2], in the camera-ready draft.
>
> We would greatly appreciate your additional feedback on whether our response has sufficiently addressed your concerns.

---

> > ### Comment · Reviewer_aKS5 · 2022-11-28
> > **Response**
> >
> > I thank the authors for their response to the review.
> > My review had two major points which I used to decide on the score:
> > * The representation of related work
> > * The readability of the theoretical analysis
> >
> > First, while the paper has cited quite a few others, I believe the contextualization necessary to appreciate and quantify the contribution of this paper is still missing.
> > To quantify what I expect here, refer to reviewer CKyn's review, where they point out that the individual components in the proposed objective (Equation 14) have been studied before.
> > Clearly discussing what aspects previous work has studied and how this work differs will strengthen and clarify the contributions of this work.
> >
> > And second, it seems all the other reviewers share my concern about the narrative and presentation of the theoretical results.
> >
> > These two areas still need significant overhaul, and I am disinclined to change my score at this time.

---

> > > ### Author Response · Authors · 2022-11-28
> > > **Thank you for your response!**
> > >
> > > Dear Reviewer aKS5,
> > >
> > > Thank you for your response!
> > >
> > > The Paper Authors have fundamental disagreements with Reviewer CKyn about related works and many other issues. Notably, **Reviewer CKyn could not point out which specific related work presented the graphical model analysis in Section 3, or where the analysis of f-divergence term in the objective was studied before.** In fact, Reviewer CKyn questioned whether the study of the f-divergence term was even necessary in the first place. **Regarding the goal likelihood term, our analysis of the term is clearly novel** when compared to any of the references listed by Reviewer CKyn (many of which we have already cited). For all three individual terms, we presented analysis that is novel and non-trivial.
> > >
> > > **While some individual components in the proposed objective have been studied before, none of those prior works have combined and analyzed them in a way that explains what the simple HER reward was optimizing under hindsight relabeling.** This is a significant difference. In fact, the prior work C-Learning paper went as far as concluding that, under the HER reward, *"hindsight relabeling produces, at best, a mediocre solution for an unclear objective function"*. Our work shows that the conclusions drawn in those prior work are not true; under our novel analysis of the individual components and joint optimization, we have managed to show that: hindsight relabeling does produce a clear objective function, and that HER reward is well-defined even for environments with continuous states.
> > >
> > > Regarding the concern that some parts of the paper are dense and difficult to follow: we strongly feel that **these difficulties stem almost entirely from the scope of the paper** – bridging together notations and ideas from three separate areas of learning to derive a divergence minimization framework for HER required a large amount of machinery. The paper draws ideas from goal-conditioned RL, inverse RL, and energy-based models, so the framework is intrinsically complex. In addition, **we need to sufficiently contextualize our work in each of those sub-fields to illustrate our novelty** (as evidenced by our discussion here), **which inevitably makes the writing dense.** We have responded to each reviewer's individual concerns, and we are still waiting for two reviewers' responses.
> > >
> > > We hope that our response above has clarified your remaining concerns about the paper.

---

> > > > ### Comment · Reviewer_aKS5 · 2022-11-28
> > > > **Thanks**
> > > >
> > > > I thank the authors for this detailed response. The above contextualization is exactly what I was hoping for in the paper! As it stands currently, this relation of the paper to previous work is not clear in the text.
> > > >
> > > > I will take this response into consideration when making my final recommendation.

---

### Decision · Program_Chairs · 2023-01-20

**Decision:**

Reject

**Justification For Why Not Higher Score:**

Too many issues compounding across literature review, experimental setup and theoretical justifications.

**Justification For Why Not Lower Score:**

N/A

**Metareview: Summary, Strengths And Weaknesses:**

This paper explores the connection between hindsight goal relabeling and imitation learning. The original submission strongly divided reviewers leading to extensive discussion. However, after further discussion of the authors' responses, all reviewers concluded that the paper was not yet ready for publication. The consensus being that there is merit to the ideas, but the current paper does not fairly present the ideas in relation to prior work, clarify important experimental choices (including outstanding questions on the differences in behaviors learnt and reward mechanism chosen) or sufficiently justify theoretical choices made (e.g. flipping the divergence for Lemma 4.1.)